# High-dose vitamin D₃ to improve outcomes in the convalescent phase of complicated severe acute malnutrition in Pakistan: a double-blind randomised controlled trial (ViDiSAM)

We have previously shown that high-dose vitamin D₃ improved weight gain and neurodevelopmental indices in children receiving standard therapy for uncomplicated severe acute malnutrition (SAM). Here we present results of a randomised placebo-controlled trial in Lahore, Pakistan, to determine whether two oral doses of 200,000 international units (IU) vitamin D₃ (the first administered on or before the day of hospital discharge and the second administered 14 days later) would benefit children aged 6-59 months during the convalescent phase of complicated SAM. Eligible participants were individually randomised to intervention vs. control arms with a one-to-one allocation ratio and stratification by hospital of recruitment using computer-generated random sequences. Double-blinding to treatment allocation was maintained by concealing allocation from participants' parents or guardians, their medical care providers, and all trial staff. The primary outcome was mean weight-for-height or -length z-score (WHZ) at 2-month follow-up. Secondary efficacy outcomes included mean WHZ at 6-month follow-up and mean lean mass index, Malawi Development Assessment Tool (MDAT) scores and serum 25-hydroxyvitamin D (25[OH]D) concentrations at 2- and 6-month follow-up. The trial has now completed. 259 children were randomised (128 to vitamin D, 131 to placebo), of whom 251 (96.9%) contributed data to analysis of the primary outcome (123 allocated to vitamin D, 128 to placebo). At 2-month follow-up, participants allocated to vitamin D had significantly higher mean serum 25(OH)D concentrations than those allocated to placebo (adjusted mean difference [aMD] 100.0 nmol/L, 95% confidence interval [CI] 72.2–127.8 nmol/L). This was not associated with an inter-arm difference in mean WHZ at 2-month follow-up (aMD 0.02, 95% CI −0.20 to 0.23), or in any anthropometric or neurodevelopmental secondary outcome assessed at 2- or 6-month follow-up. The intervention was safe. In conclusion, high-dose vitamin D₃ elevated mean

✉e-mail: javeria.hasan@hotmail.com; a.martineau@qmul.ac.uk

serum 25(OH)D concentrations in children receiving standard therapy for complicated SAM in Pakistan, but did not influence any anthropometric or neurodevelopmental outcome studied. The trial was registered at Clinical-Trials.gov with the identifier NCT04270643.

An estimated 19 million children globally suffer from the most critical form of undernutrition, severe acute malnutrition (SAM)[1], of whom around 1.4 million live in Pakistan[2]. The condition is characterised by loss of muscle and fat tissue, associated with increased systemic inflammation and susceptibility to infections. SAM is classified as complicated (~20% of cases) or uncomplicated (~80%) according to the presence or absence of medical complications such as acute lower respiratory infection, severe dehydration, severe anaemia, severe pitting oedema, anorexia, hypothermia, hyperpyrexia or hypoglycaemia. Children hospitalised with complicated SAM have high mortality (10-30%); those who survive are at increased risk of post-discharge mortality and readmission to hospital over the subsequent year[3,4] and long-term adverse effects on their physical and cognitive development[5,6], which may compromise their economic productivity as adults[7].

Initial treatment for complicated SAM involves stabilisation as a hospital in-patient with management of immediate complications, provision of low-protein milk-based feeds, and administration of broad-spectrum parenteral antibiotics. Once appetite has been regained, the child transitions to ready-to-use therapeutic food (RUTF) —an energy-dense micronutrient-enriched paste. This represents the mainstay for on-going treatment, which is continued in the community following hospital discharge until the child gains appropriate weight and any oedema has resolved for at least 2 weeks.

There have been no new interventions for complicated SAM since WHO guidelines were published in 1999. The WHO has highlighted the need for research to identify adjunctive therapies that may improve response to RUTF, including administration of broad-spectrum antibiotics and high-dose vitamin A[1]. Vitamin D has been shown to have favourable effects on skeletal muscle function[8] and neurodevelopment[9] as well as boosting antimicrobial immune function and accelerating resolution of systemic inflammatory responses[10,11]. These effects might exert benefits in a condition where increased systemic inflammation and susceptibility to infections are associated with adverse outcomes[12,13]. The case for investigating vitamin D as an adjunct to existing treatments for SAM is further strengthened by reports from observational studies that rickets and vitamin D deficiency are common in children with SAM[14–16] and that vitamin D deficiency associates with severe wasting in malnourished children[17].

We previously conducted a community-based randomised controlled trial (RCT) in children with uncomplicated SAM in Lahore, Pakistan, which showed that administration of two oral doses of 200,000 IU vitamin D3 alongside RUTF elevated serum 25(OH)D concentrations into the high physiological range and improved weight gain and neurodevelopmental indices over the initial 8 weeks of treatment[18]. These findings prompted us to conduct a second RCT in the same setting, designed to determine whether this intervention might also influence weight gain, body composition or neurodevelopmental indices in children recovering from complicated SAM. We chose to mirror the protocol of our previous trial by intervening at the point of hospital discharge rather than at the point-of hospital admission, in order to ameliorate pathology that persists beyond the period of hospitalisation[19] and to address the evidence gap in improving convalescent care, highlighted by the latest WHO Guidelines for the management of SAM[20]. We used the Malawi Development Assessment Tool (MDAT) to assess neurodevelopmental status, as this instrument has been used across multiple settings, including Pakistan[21], and works well across the age-range of children eligible to

participate in this study as a directly observed assessment. Moreover, the MDAT yields a continuous read-out, across multiple domains, that is more sensitive for detection of an effect of the intervention than the Denver Developmental Screening Test that we used in our previous trial[18].

In this work, we show that high-dose vitamin D3 is effective in elevating serum 25(OH)D concentrations in children receiving standard therapy for complicated SAM in Pakistan, but that this is not associated with any effect on anthropometric or neurodevelopmental outcomes investigated.

## Results

### Participants

579 children were screened for eligibility from December 2021 to February 2023. Dates of first and last enrolments were 19 January 2022 and 28 February 2023, respectively. Parental written informed consent was obtained for 273 potentially eligible children and a blood sample was collected to screen for baseline hypercalcaemia. 259 participants (94.9%) were confirmed eligible and randomly assigned to receive vitamin D (n = 128) or placebo (n = 131; Fig. 1). Reasons for ineligibility

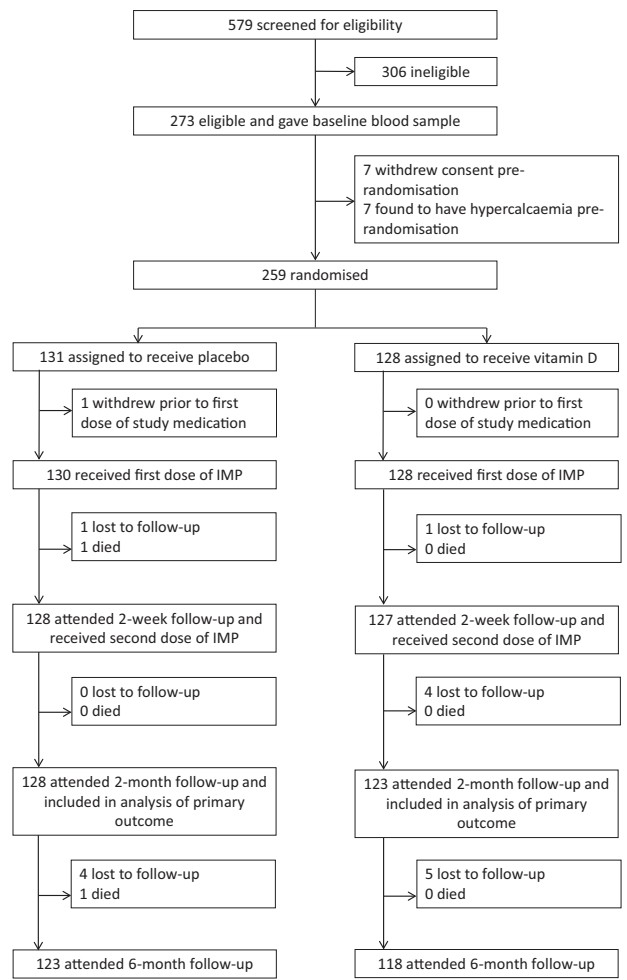

**Fig. 1 | Trial profile.**

are provided in Table S1, Supplementary Information. Participants' baseline characteristics are presented in Table 1. Overall, mean age at enrolment was 14.5 months (standard deviation [SD] 8.2 months), 56.8% of participants were female and 81.5% were of Punjabi ethnic origin. Mean baseline serum 25(OH)D concentration was 54.4 nmol/L (SD 47.3 nmol/L). Participants' baseline characteristics were well balanced between those allocated to receive vitamin D vs. placebo. 255 (98.5%) randomised participants received both doses of IMP as assigned (Table S2, Supplementary Information), and 251 (96.9%) attended 2-month follow-up and contributed data to analysis of the primary outcome (Fig. 1). The trial ended when the final participant completed 6-month follow-up.

## Efficacy outcomes

Overall, allocation to vitamin D vs. placebo did not influence the primary outcome of WHZ at 2-month follow-up (adjusted mean difference [aMD] 0.02, 95% confidence interval [CI] −0.20 to 0.23, $P = 0.89$), or any other anthropometric or neurodevelopmental outcome assessed at 2- or 6-month follow-up (Table 2). Participants allocated to vitamin D had higher mean serum total 25(OH)D concentrations, higher mean serum total 24 R,25(OH)$_2$D concentrations and lower 25(OH)D-to-24R,25(OH)$_2$D ratios than those allocated to placebo at both 2- and 6-month follow-up (Table 2 and Fig. 2). No inter-arm differences were seen for other serum biochemical outcomes, full blood count parameters or faecal inflammatory markers at either 2- or 6-month follow-up (Table 2). Pre-specified sub-group analyses comparing the effect of allocation in participants who had baseline WHZ ≥ −3.0 vs. < −3.0 yielded some evidence of effect modification on the outcome of mean WHZ at 2-month follow-up (with the latter group benefitting more from the intervention, $P$ for interaction 0.015), but not for other outcomes investigated (Table S3, Supplementary Information). $P$ values for interaction for all sub-group analyses by sex were >0.05 (Table S4, Supplementary Information). Sub-group analyses by baseline vitamin D status were conducted in a subset of 106 participants for whom both baseline serum 25(OH)D concentrations and outcome measures were available: $P$ values for interaction for all of these sub-group analyses were also >0.05 (Table S5, Supplementary Information). Baseline characteristics of participants included in this sub-group analysis were similar to those of participants overall (Table S6, Supplementary Information).

## Safety outcomes

Table 3 presents details of safety outcomes by allocation. Two children died during follow-up (both allocated to placebo), and 12 others experienced non-fatal serious adverse events (3 in the vitamin D arm and 9 in the placebo arm; for details of individual serious adverse events, see Table S7, Supplementary Information). Seven children suffered a relapse of SAM during follow-up (3 vs. 4 allocated to vitamin D vs placebo, respectively). Hypercalcaemia was detected in seven children (2 vs. 5 allocated to vitamin D vs. placebo, respectively), hypervitaminosis D in 23 (22 vs. 1 allocated to vitamin D vs. placebo, respectively), and hypercalciuria in 94 (44 vs. 50 allocated to vitamin D vs. placebo, respectively; Table 3).

## Discussion

We present findings of a randomised controlled trial investigating effects of high-dose oral vitamin D supplementation in children in the convalescent phase of complicated SAM, which remains a high-risk condition with unacceptably poor outcomes and few innovations in management over the past three decades. Administration of two oral bolus doses of 200,000 IU vitamin D$_3$ (the first administered on or before the day of hospital discharge and the second administered 14 days later) was effective in elevating mean serum 25(OH)D concentrations at 2- and 6-month follow-up, but it did not influence the primary outcome of WHZ overall, or any other anthropometric or neurodevelopmental outcome investigated.

Null findings from this study contrast with those of our previous trial in Lahore, in which we reported that the same intervention improved weight gain and neurodevelopmental indices in children receiving standard therapy for uncomplicated SAM in the community[18]. This difference in findings may reflect the fact that participants in the current trial had complicated (as opposed to uncomplicated) SAM: effects of the intervention may have been masked by concurrent administration of more potent pharmacological therapies, or greater disease severity (including infections in most children at baseline) may have been more resistant to amelioration by a micronutrient supplement. Alternatively, absorption of this orally administered fat-soluble vitamin may have been compromised in children with complicated vs. uncomplicated SAM, in whom maldigestion and malabsorption of dietary lipid have been reported[22]. Low 25(OH)D concentrations following vitamin D supplementation might also reflect dysregulated vitamin D metabolism, which has been reported to occur in other inflammatory conditions[23-25].

Our study also showed that the intervention was safe. Specifically, administration of vitamin D was not associated with increased incidence of hypercalcaemia or hypercalciuria despite the very high dose given, which elevated 25(OH)D concentrations above 220 nmol/L in more than 20 participants. Hypercalciuria was common at baseline and at follow-up among participants in both study arms: however, we highlight that this was based on testing of 'spot urines': further studies using 24-hour urine collection are needed to establish whether hypercalciuria is indeed a clinical feature of complicated SAM.

Our trial had several strengths. The study was double-blind and placebo-controlled, and administration of IMP was directly observed, ensuring near-perfect adherence. Furthermore, participant retention was higher than anticipated in our sample size calculation, maximising power and internal validity. Prevalence of vitamin D deficiency (defined as serum 25[OH]D concentration <50 nmol/L) among study participants at baseline (63.4%) was similar to that reported among children aged 6-59 months in the 2018 Pakistan National Nutritional Survey (62.7%)[26], supporting the external validity of our findings. We investigated effects of the intervention on a very broad range of anthropometric, neurodevelopmental, biochemical, haematological and safety outcomes, assessed using validated tools with continuous read-outs such as bioimpedance analysis and MDAT to maximise our sensitivity for signal detection. The dose of vitamin D administered was effective in elevating mean serum 25(OH)D concentrations. However, there is some debate regarding the relative merits of intermittent bolus dosing regimens (such as the one we used) vs. daily administration of vitamin D[27-29]. The former approach has been shown to result in high circulating concentrations of 24R,25(OH)$_2$D with plateauing of 1,25(OH)$_2$D concentrations, with the result that 24R,25(OH)$_2$D may antagonise vitamin D signalling and attenuate potential therapeutic effects of supplementation[30]. Given that we showed sustained elevation of serum 24R,25(OH)$_2$D concentrations and reduction in 25(OH)D:24R,25(OH)$_2$D ratios among participants randomised to intervention, it may be that smaller regular doses of vitamin D could have exerted more benefit by elevating serum 25(OH)D and 1,25(OH)$_2$D concentrations without causing large increases in 24R,25(OH)$_2$D. However, the fact that the same intervention showed benefit in children with uncomplicated SAM[18] would argue against this hypothesis.

Our study also had some limitations. All participants had non-oedematous SAM, so our findings cannot be generalised to children with oedematous SAM. Although we recruited at two centres, both were in the same geographical location, so results are not generalisable to other settings e.g. in Africa. Findings cannot be generalised to children with severe disability, as they would have been excluded based on inability to complete the MDAT assessment at baseline. Three children randomised to the placebo arm of the trial had 25(OH)D

**Table 1 | Baseline participant characteristics, by allocation and overall**

| | | Placebo (*n* = 131) | Vitamin D (*n* = 128) | Overall (*n* = 259) |
|---|---|---|---|---|
| Sociodemographic[a] | | | | |
| Mean age, months (s.d.) [range] | | 15.1 (8.4) [6.0–48.1] | 14.0 (8.0) [6.0–49.7] | 14.5 (8.2) [6.0–49.7] |
| Sex, *n* (%) | Female | 74 (56.5%) | 73 (57.0%) | 147 (56.8%) |
| | Male | 57 (43.5%) | 55 (43.0%) | 112 (43.2%) |
| Ethnic origin, *n* (%) | Kashmiri | 1 (0.8%) | 0 (0.0%) | 1 (0.4%) |
| | Muhajir | 2 (1.5%) | 3 (2.3%) | 5 (1.9%) |
| | Pashtoon | 4 (3.1%) | 6 (4.7%) | 10 (3.9%) |
| | Punjabi | 109 (83.2%) | 102 (79.7%) | 211 (81.5%) |
| | Saraiki | 0 (0.0%) | 2 (1.6%) | 2 (0.8%) |
| | Other | 15 (11.5%) | 15 (11.7%) | 30 (11.6%) |
| Study site, *n* (%) | Sir Ganga Ram Hospital | 64 (48.9%) | 65 (50.8%) | 129 (49.8%) |
| | THQ Hospital | 67 (51.1%) | 63 (49.2%) | 130 (50.2%) |
| Number of younger siblings at home, *n* (%) | 0 | 122 (93.1%) | 115 (90.6%) | 237 (91.9%) |
| | 1 | 7 (5.3%) | 11 (8.7%) | 18 (7.0%) |
| | 2 or more | 2 (1.5%) | 1 (0.8%) | 3 (1.2%) |
| Number of older siblings at home, *n* (%) | 0 | 19 (14.5%) | 23 (18.0%) | 42 (16.2%) |
| | 1 | 37 (28.2%) | 37 (28.9%) | 74 (28.6%) |
| | 2 | 36 (27.5%) | 21 (16.4%) | 57 (22.0%) |
| | 3 | 17 (13.0%) | 20 (15.6%) | 37 (14.3%) |
| | 4 | 13 (9.9%) | 18 (14.1%) | 31 (12.0%) |
| | 5 or more | 9 (6.9%) | 9 (7.0%) | 18 (6.9%) |
| Mother's education, *n* (%) | No formal schooling | 47 (35.9%) | 45 (35.2%) | 92 (35.5%) |
| | Primary education only | 50 (38.2%) | 41 (32.0%) | 91 (35.1%) |
| | Secondary education and above | 34 (26.0%) | 42 (32.8%) | 76 (29.3%) |
| Father's education, *n* (%) | No formal schooling | 52 (39.7%) | 44 (34.4%) | 96 (37.1%) |
| | Primary education only | 42 (32.1%) | 43 (33.6%) | 85 (32.8%) |
| | Secondary education and above | 37 (28.2%) | 41 (32.0%) | 78 (30.1%) |
| Mean monthly household income, PKR (s.d.) | | 21359 (13475) | 23820 (14680) | 22575 (14110) |
| Type of residence | Cottage | 0 (0.0%) | 1 (0.8%) | 1 (0.4%) |
| | Flat/apartment | 113 (86.3%) | 114 (89.1%) | 227 (87.6%) |
| | Terraced house | 10 (7.6%) | 12 (9.4%) | 22 (8.5%) |
| | Other | 8 (6.1%) | 1 (0.8%) | 9 (3.5%) |
| Vaccination status, *n* (%) | Up-to-date | 94 (71.8%) | 84 (65.6%) | 178 (68.7%) |
| | Not up-to-date | 37 (28.2%) | 44 (34.4%) | 81 (31.3%) |
| Early life characteristics | | | | |
| Gestational age, *n* (%) | Term | 121 (92.4%) | 112 (87.5%) | 233 (90.0%) |
| | Pre-term | 10 (7.6%) | 16 (12.5%) | 26 (10.0%) |
| Child's birthweight, *n* (%) | Very low birth weight (<1.5 kg) | 1 (0.8%) | 1 (0.8%) | 2 (0.8%) |
| | Low birth weight (1.5 to <2.5 kg) | 52 (39.7%) | 56 (43.8%) | 108 (41.7%) |
| | Normal (≥2.5 kg <4.0 kg) | 76 (58.0%) | 70 (54.7%) | 146 (56.4%) |
| | Too heavy (≥4.0 kg) | 2 (1.5%) | 1 (0.8%) | 3 (1.2%) |
| Breastfeeding, *n* (%) | Still breastfeeding | 46 (35.1%) | 46 (35.9%) | 92 (35.5%) |
| | Breastfed for 0–5 months then stopped | 55 (42.0%) | 60 (46.9%) | 115 (44.4%) |
| | Breastfed for 6–12 months then stopped | 12 (9.2%) | 10 (7.8%) | 22 (8.5%) |
| | Breastfed >12 months then stopped | 18 (13.7%) | 12 (9.4%) | 30 (11.6%) |
| Age at weaning, months, *n* (%) | Still exclusively breastfeeding | 0 (0.0%) | 3 (2.3%) | 3 (1.2%) |
| | <4 months | 4 (3.1%) | 4 (3.1%) | 8 (3.1%) |
| | 4–5 months | 29 (22.1%) | 38 (29.7%) | 67 (25.9%) |
| | 6–8 months | 79 (60.3%) | 72 (56.2%) | 151 (58.3%) |
| | 9–11 months | 15 (11.5%) | 6 (4.7%) | 21 (8.1%) |
| | 12–23 months | 4 (3.1%) | 5 (3.9%) | 9 (3.5%) |

**Table 1 (continued) | Baseline participant characteristics, by allocation and overall**

| | | Placebo (*n* = 131) | Vitamin D (*n* = 128) | Overall (*n* = 259) |
|---|---|---|---|---|
| **Anthropometry at baseline** | | | | |
| Weight-for-height or -length z-score (s.d.) [range] | | −3.5 (0.8) [−5.9 to −1.8] | −3.5 (1.0) [−6.4 to −1.0] | −3.5 (0.9) [−6.4 to −1.0] |
| Mean mid-upper arm circumference, cm (s.d.) | | 10.5 (1.0) | 10.5 (0.9) | 10.5 (0.9) |
| Bilateral pitting oedema at hospital admission, *n* (%) | Present | 0 (0%) | 0 (0%) | 0 (0%) |
| | Absent | 131 (100.0%) | 128 (100.0%) | 259 (100.0%) |
| Mean weight-for-age z-score (s.d.) | | −4.2 (1.0) | −4.2 (1.1) | −4.2 (1.0) |
| Mean height/length-for-age z-score (s.d.) | | −3.5 (0.8) | −3.5 (1.0) | −3.5 (0.9) |
| Mean head circumference z-score (s.d.) | | −2.8 (1.3) | −2.8 (1.2) | −2.8 (1.2) |
| Lean mass index, $\Omega^{-1}$ (s.d.) | | 12.0 (2.1) | 11.9 (2.0) | 11.9 (2.0) |
| Complication present at hospital admission, n (%) | Acute lower respiratory infection | 27 (20.6%) | 24 (18.8%) | 51 (19.7%) |
| | Acute upper respiratory infection | 10 (7.6%) | 9 (7.0%) | 19 (7.3%) |
| | Anorexia | 5 (3.8%) | 3 (2.3%) | 8 (3.1%) |
| | Gastroenteritis/Diarrhoea | 44 (33.6%) | 60 (46.9%) | 104 (40.2%) |
| | Hyperpyrexia | 5 (3.8%) | 1 (0.8%) | 6 (2.3%) |
| | Hypoglycaemia | 1 (0.8%) | 0 (0.0%) | 1 (0.4%) |
| | Severe anaemia[b] | 11 (8.4%) | 5 (3.9%) | 16 (6.2%) |
| | Severe dehydration | 22 (16.8%) | 22 (17.2%) | 44 (17.0%) |
| | Other | 6 (4.6%) | 4 (3.1%) | 10 (3.9%) |
| **Neurodevelopmental status at baseline**[c] | | | | |
| Mean overall MDAT score (all domains) (s.d.) | | 51.4 (21.3) | 50.3 (20.8) | 50.8 (21.0) |
| Mean Gross motor score (s.d.) | | 13.7 (5.4) | 13.5 (5.3) | 13.6 (5.3) |
| Mean Fine motor score (s.d.) | | 14.1 (5.9) | 13.7 (5.7) | 13.9 (5.8) |
| Mean Language score (s.d.) | | 10.6 (5.7) | 10.2 (5.3) | 10.4 (5.5) |
| Mean Social score (s.d.) | | 13.0 (5.3) | 12.8 (5.5) | 12.9 (5.4) |
| **Blood biochemistry**[d] | | | | |
| Mean total serum 25(OH)D, nmol/L (s.d.) [range] | | 51.8 (45.9) [6.1–250.5] | 57.1 (48.9) [6.1–292.4] | 54.4 (47.3) [6.1–292.4] |
| Total serum 25(OH)D category | <25 nmol/L, *n* (%) | 15 (27.8%) | 10 (18.9%) | 25 (23.4%) |
| | ≥25 nmol/L and <50 nmol/L, *n* (%) | 22 (40.7%) | 21 (39.6%) | 43 (40.2%) |
| | ≥50 nmol/L and <75 nmol/L, *n* (%) | 7 (13.0%) | 12 (22.6%) | 19 (17.8%) |
| | ≥75 nmol/L, *n* (%) | 10 (18.5%) | 10 (18.9%) | 20 (18.7%) |
| Mean total serum 24 R,25(OH)$_2$D, nmol/L (s.d.) | | 2.9 (4.4) | 3.4 (5.0) | 3.1 (4.7) |
| Mean total 25(OH)D: total 24 R,25(OH)$_2$D ratio (s.d.) | | 27.2 (9.8) | 24.9 (11.7) | 26.1 (10.8) |
| Mean serum albumin, g/L (s.d.) | | 38.0 (5.5) | 37.4 (5.5) | 37.7 (5.5) |
| Mean serum C-reactive protein, mg/L (s.d.) | | 4.8 (11.4) | 5.5 (9.2) | 5.1 (10.3) |
| Mean serum albumin-adjusted calcium, mmol/L (s.d.)[e] | | 2.36 (0.19) | 2.40 (0.16) | 2.38 (0.18) |
| Mean serum total alkaline phosphatase, IU/L (s.d.) | | 317.6 (358.6) | 238.6 (93.6) | 278.5 (264.9) |
| Mean serum PTH, pmol/L (s.d.) | | 4.3 (7.4) | 2.3 (2.9) | 3.3 (5.7) |
| Mean serum ferritin, µg/L (s.d.) | | 45.0 (75.7) | 62.8 (90.7) | 53.8 (83.5) |
| Mean serum hepcidin, ng/ml (s.d.) | | 17.1 (35.6) | 23.4 (42.5) | 20.2 (39.1) |
| **Urine biochemistry** | | | | |
| Mean urinary calcium creatinine molar ratio (s.d.) | | 1.85 (6.06) | 1.77 (5.44) | 1.81 (5.75) |
| Mean urinary osmolality, mOsm/kg (s.d.) | | 612.34 (275.95) | 670.39 (302.79) | 641.03 (290.42) |
| Proportion with urinary calcium:creatinine molar ratio >1.0 | | 39/131 (29.8%) | 32/128 (25.0%) | 71/259 (27.4%) |
| **Full blood count parameters**[f] | | | | |
| Mean haemoglobin, g/dL (s.d.) | | 8.8 (1.9) | 9.3 (1.8) | 9.0 (1.9) |
| Mean corpuscular volume, fL (s.d.) | | 69.9 (13.8) | 69.6 (12.0) | 69.8 (12.9) |
| Mean corpuscular haemoglobin concentration, g/dL (s.d.) | | 21.2 (5.3) | 21.4 (4.5) | 21.3 (4.9) |
| Mean total WBC, ×10$^9$/L (s.d.) | | 12.5 (5.4) | 12.5 (6.1) | 12.5 (5.7) |
| Mean neutrophil count, ×10$^9$/L (s.d.) | | 4.0 (3.2) | 3.7 (2.0) | 3.9 (2.7) |

**Table 1 (continued) | Baseline participant characteristics, by allocation and overall**

| | Placebo (*n* = 131) | Vitamin D (*n* = 128) | Overall (*n* = 259) |
|---|---|---|---|
| Mean lymphocyte count, ×10⁹/L (s.d.) | 6.9 (3.1) | 6.8 (3.2) | 6.9 (3.2) |
| Mean monocyte count, ×10⁹/L (s.d.) | 1.1 (0.8) | 1.0 (0.5) | 1.0 (0.7) |
| Mean platelet count, ×10⁹/L (s.d.) | 453 (193) | 457 (191) | 455 (192) |
| Faecal inflammatory markers | | | |
| Median faecal myeloperoxidase, ng/ml (IQR) | 1793.0 (766.9–5342.0) | 2199.8 (954.5–5365.1) | 2023.4 813.6–5342.0) |
| Median faecal neopterin, nmol/L (IQR) | 1249.9 (449.4–2938.7) | 1431.8 (571.7–2946.8) | 1315.4 (545.8–2938.7) |
| Median faecal alpha-1 antitrypsin, mg/dL (IQR) | 11.8 (6.2–22.3) | 13.4 (6.3–22.3) | 12.6 (6.2–22.3) |

*THQ Hospital* Tehsil Headquarter Hospital, *PKR* Pakistani Rupee, *KHz* Kilohertz, *MDAT* Malawi Developmental Assessment Tool, *WBC* white blood cell, *PTH* parathyroid hormone, *25(OH)D* 25-hydroxyvitamin D, *S.d.* standard deviation, *CI* confidence interval.

[a]Sociodemographic data missing for one participant in the vitamin D arm.
[b]Severe anaemia defined as haemoglobin concentration <6 g/dL.
[c]Neurodevelopmental data missing for one participant in both the vitamin D arm and placebo arms.
[d]Blood biochemistry data missing for 83 participants in both the vitamin D and placebo arms.
[e]Adjusted calcium (mmol/L) = measured total calcium (mmol/L) + 0.02 × (40 − serum albumin [g/L]).
[f]Haematological data missing for one participant in the placebo arm.

concentrations >150 nmol/L at baseline, suggesting that they might have taken high-dose vitamin D supplements not long before enrolment. However, the small number of such participants provides some reassurance that contamination bias was not responsible for our null findings. Finally, serum samples were only available for biochemical analysis for a subset of participants, which limited power for sub-group analyses by baseline vitamin D status.

In conclusion, we found that high-dose vitamin $D_3$ safely elevated mean serum 25(OH)D concentrations in children receiving standard therapy for complicated SAM in Pakistan, but did not influence any anthropometric or neurodevelopmental outcome overall. These results contrast with previous findings in children with uncomplicated SAM, studied in the same setting.

## Methods
### Trial approvals, registration, design, setting, sponsorship and monitoring
Trial conduct complied with all relevant ethical regulations. The study was approved by Research Ethics Committees at the Sir Ganga Ram Hospital (ref 62.Preventive Paeds/Trial PU/ERC) and the Tehsil Headquarter Hospital (ref IRD_IRB_2020_11_007), the National Bioethics Committee of Pakistan (ref. 4–87/NBC-516/20/477) and the Liverpool School of Tropical Medicine Research Ethics Committee (ref. 20-028) and registered on ClinicalTrials.gov (ref. NCT04270643). We conducted a phase II, double-blind, parallel-group individually randomised placebo-controlled trial. Participants were recruited from the Sir Ganga Ram Hospital and the Tehsil Headquarter Hospital, in Lahore, Pakistan (latitude 31.5° N), and were followed-up in the Outpatient Centre or community settings in the surrounding area. The trial was sponsored by Queen Mary University of London., Trial monitoring was performed by DRK Pharma Solutions (Lahore, Pakistan).

### Participants
Inclusion criteria were written informed consent of the parent or guardian; participant age of 6–59 months at enrolment; a diagnosis of complicated SAM at the point-of hospital admission, as defined by the World Health Organisation (WHO)[31]; and planned discharge from in-patient care. Exclusion criteria were ingestion of a dose of vitamin D > 200,000 IU (5 mg) in the last 3 months; known diagnosis of primary hyperparathyroidism or sarcoidosis (i.e. conditions predisposing to vitamin D hypersensitivity); known neurodevelopmental disorder (e.g. cerebral palsy); HIV infection; taking anti-tuberculosis treatment; inability to assess a child's developmental status at baseline using the MDAT[32]; clinical signs of rickets; child's family planning to move out of the study area within 6 months of enrolment; and baseline albumin-adjusted serum calcium concentration >2.65 mmol/L.

### Enrolment
Parents or legal guardians were invited to provide written informed consent for their child to participate in the trial. Those who accepted were asked to provide information on children's social circumstances and medical history, which were captured on an electronic case report form. A physical examination was also performed to assess children for signs of rickets (genu varum, genu valgum, windswept deformity of the knees, rickety rosary, frontal bossing or wrist-widening). Children exhibiting such signs were excluded from the trial and referred to the nearest appropriate clinic for investigation and treatment. Children without rickets underwent assessment of oedema and anthropometric measurements for mid-upper arm circumference, head circumference, weight and height (or length if unable to stand). The child's body composition was also measured by bioimpedance analysis using the Bodystat 500 MDD system (Bodystat Ltd, Douglas, Isle of Man, UK) and a baseline developmental assessment was performed using the MDAT, both as previously described[32,33]. Finally, if the developmental assessment was successfully completed, and if results of a routine HIV test performed as part of standard clinical care were unavailable, a point-of-care HIV test was performed. If this test was positive, the child was excluded from the trial and referred to the appropriate care team for confirmatory testing and further management as appropriate. If the HIV test was negative, a 5 ml blood sample and a 5 ml urine sample were collected to measure baseline biochemical and haematological markers as described below. A stool sample was also collected for measurement of baseline faecal inflammatory markers.

### Randomisation and blinding
Eligible participants were individually randomised to intervention vs. control arms with a one-to-one allocation ratio and stratification by hospital of recruitment using computer-generated random sequences; a full description of how this was done is presented in Supplementary Information. Treatment allocation was concealed from participants' parents or guardians, their medical care providers, and all trial staff (including senior investigators and those assessing outcomes) so that the double-blind was maintained.

### Intervention
IMP comprised two ampoules of a solution manufactured by GT Pharma Ltd containing either 5 mg (200,000 IU) vitamin $D_3$ in 1 ml ethyl oleate, or 1 ml ethyl oleate solution with no vitamin $D_3$. Active and placebo ampoules had identical appearance, and their contents had identical appearance and taste. These contents were drawn up into a syringe and administered orally, by study staff, with the first dose administered on or before the day of the participant's hospital

**Table 2 | Efficacy outcomes by allocation**

| | Timepoint, months from baseline | Placebo: mean value (s.d.), [n] | Vitamin D: mean value (s.d.), [n] | Adjusted mean difference (95% CI)[a] |
|---|---|---|---|---|
| **Anthropometric outcomes** | | | | |
| Weight-for-height or -length z-score | +2 | −2.59 (1.09), [128] | −2.58 (1.10), [123] | 0.02 (−0.20, 0.23)[b] |
| | +6 | −2.18 (1.07), [123] | −2.17 (1.09), [118] | 0.01 (−0.22, 0.24) |
| Height- or length-for-age z-score | +2 | −3.12 (1.44), [128] | −3.03 (1.47), [123] | −0.00 (−0.13, 0.13) |
| | +6 | −3.00 (1.37), [123] | −2.97 (1.41), [118] | −0.05 (−0.25, 0.15) |
| Weight-for-age z-score | +2 | −3.50 (1.11), [128] | −3.47 (1.11), [123] | 0.01 (−0.16, 0.18) |
| | +6 | −3.09 (1.09), [123] | −3.08 (1.06), [118] | −0.00 (−0.22, 0.22) |
| Head circumference-for-age z-score | +2 | −2.45 (1.17), [128] | −2.52 (1.16), [123] | −0.00 (−0.15, 0.14) |
| | +6 | −2.19 (1.12), [123] | −2.26 (1.09), [118] | −0.08 (−0.27, 0.12) |
| Mid-upper arm circumference, cm | +2 | 11.57 (0.91), [128] | 11.65 (0.89), [123] | 0.07 (-0.12, 0.26) |
| | +6 | 12.27 (0.82), [123] | 12.26 (0.78), [118] | −0.00 (−0.22, 0.21) |
| Lean mass index, $\Omega^{-1}$ | +2 | 12.61 (2.11), [128] | 12.65 (2.36), [123] | 0.08 (−0.40, 0.56) |
| | +6 | 12.75 (2.22), [123] | 12.35 (1.90), [118] | −0.31 (−0.79, 0.17) |
| **Neurodevelopmental outcomes** | | | | |
| Overall MDAT score | +2 | 62.33 (22.36), [128] | 60.68 (21.88), [123] | −0.41 (−1.67, 0.85) |
| | +6 | 77.01 (21.08), [123] | 76.00 (20.86), [118] | 0.18 (−1.41, 1.77) |
| Gross motor score | +2 | 16.49 (5.83), [128] | 16.22 (5.73), [123] | −0.09 (−0.61, 0.43) |
| | +6 | 20.53 (6.00), [123] | 20.31 (5.58), [118] | 0.01 (−0.63, 0.66) |
| Fine motor score | +2 | 16.85 (5.89), [128] | 16.46 (5.68), [123] | 0.08 (−0.37, 0.52) |
| | +6 | 20.59 (5.77), [123] | 20.53 (5.54), [118] | 0.34 (−0.28, 0.96) |
| Language score | +2 | 12.88 (6.08), [128] | 12.37 (5.81), [123] | −0.14 (−0.56, 0.27) |
| | +6 | 15.83 (5.81), [123] | 15.54 (6.04), [118] | −0.03 (−0.59, 0.54) |
| Social score | +2 | 16.11 (5.60), [128] | 15.63 (5.59), [123] | −0.29 (−0.83, 0.25) |
| | +6 | 20.06 (5.07), [123] | 19.61 (4.80), [118] | −0.21 (−0.84, 0.41) |
| **Biochemical outcomes** | | | | |
| Mean serum total 25(OH)D, nmol/L (s.d., range) | +2 | 66.9 (44.2, 9.2–238.8), [47] | 168.0 (123.9, 8.8–511.5), [62] | 100.0 (72.2, 127.8) |
| | +6 | 36.2 (21.8, 4.2 to 127.8), [55] | 62.5 (56.1, 11.9 to 315.4), [66] | 25.1 (9.1, 41.1) |
| Mean serum total 24 R,25(OH)$_2$D, nmol/L (s.d.) | +2 | 4.3 (4.4), [47] | 12.1 (9.6), [62] | 7.65 (5.29, 10.01) |
| | +6 | 1.5 (1.5), [55] | 3.5 (4.5), [66] | 1.81 (0.37, 3.24) |
| Mean serum total 25(OH)D: total 24 R,25(OH)$_2$D ratio (s.d.) | +2 | 20.6 (8.3), [47] | 16.6 (6.7), [62] | −3.96 (−7.45, −0.48) |
| | +6 | 29.3 (8.6), [55] | 25.0 (9.4), [66] | −4.38 (−7.89, −0.87) |
| Mean serum albumin-adjusted calcium, mmol/L (s.d.) | +2 | 2.43 (0.13), [128] | 2.47 (0.21), [123] | 0.03 (−0.02, 0.07) |
| | +6 | 2.40 (0.13), [121] | 2.43 (0.08), [118] | 0.02 (−0.01, 0.06) |
| Mean serum total alkaline phosphatase, IU/L (s.d.) | +2 | 259.7 (105.5), [45] | 261.4 (136.9), [61] | −26.22 (−84.43, 31.98) |
| | +6 | 256.5 (121.1), [54] | 238.9 (71.0), [66] | −42.10 (−99.34, 15.14) |
| Mean serum PTH, pmol/L (s.d.) | +2 | 1.3 (1.4), [45] | 2.1 (2.0), [60] | 0.9 (−0.7, 2.4) |
| | +6 | 3.0 (2.9), [53] | 2.3 (1.1), [64] | 0.0 (−1.3, 1.3) |
| Mean serum C-reactive protein, mg/L (s.d.) | +2 | 5.6 (17.5), [46] | 3.8 (8.3), [62] | −1.19 (−5.09, 2.71) |
| | +6 | 5.0 (13.6), [55] | 3.6 (7.5), [66] | −1.33 (−5.10, 2.43) |
| Mean serum ferritin, μg/L (s.d.) | +2 | 36.1 (52.4), [46] | 35.9 (47.9), [62] | -0.14 (−28.24, 27.95) |
| | +6 | 24.6 (33.5), [54] | 22.3 (25.9), [65] | -2.41 (−28.09, 23.27) |
| Mean serum hepcidin, ng/ml (s.d.) | +2 | 32.7 (103.7), [46] | 25.0 (49.5), [62] | -8.38 (−33.56, 16.80) |
| | +6 | 13.9 (27.7), [53] | 13.5 (32.5), [66] | -2.72 (-19.62, 14.19) |
| Mean serum albumin, g/L (s.d.) | +2 | 40.1 (4.3), [128] | 39.8 (3.3), [122] | -0.13 (-1.25, 0.99) |
| | +6 | 40.0 (3.7), [122] | 40.3 (3.4), [118] | 0.48 (-0.69, 1.65) |
| Mean urinary calcium:creatinine molar ratio (s.d.) | +2 | 1.21 (3.36), [128] | 0.89 (1.24), [123] | -0.34 (-1.44, 0.79) |
| | +6 | 1.23 (5.08), [118] | 0.66 (0.51), [115] | -0.58 (-1.83, 0.67) |
| Mean urinary osmolality, mOsm/kg (s.d.) | +2 | 716.4 (275.5), [128] | 709.0 (295. 8), [123] | -9.69 (-78.74, 59.36) |
| | +6 | 753.1 (218.8), [118] | 799.8 (234.3), [115] | 46.14 (-20.41, 112.69) |
| **Full blood count parameters** | | | | |
| Mean haemoglobin, g/dL (s.d.) | +2 | 9.6 (1.8), [128] | 9.9 (1.5), [121] | -0.01 (−0.38, 0.36) |
| | +6 | 9.3 (2.1), [121] | 9.7 (1.7), [118] | 0.13 (−0.30, 0.56) |

**Table 2 (continued) | Efficacy outcomes by allocation**

| | Timepoint, months from baseline | Placebo: mean value (s.d.), [n] | Vitamin D: mean value (s.d.), [n] | Adjusted mean difference (95% CI)a |
|---|---|---|---|---|
| Mean corpuscular volume, fL (s.d.) | +2 | 68.6 (10.3), [128] | 69.3 (9.0), [121] | 0.56 (−1.42, 2.54) |
| | +6 | 66.1 (10.1), [121] | 66.4 (9.6), [118] | 0.25 (−2.18, 2.68) |
| Mean corpuscular haemoglobin concentration, g/dL (s.d.) | +2 | 21.2 (4.1), [128] | 21.2 (3.6), [121] | −0.11 (−0.82, 0.60) |
| | +6 | 20.1 (4.4), [121] | 20.4 (4.0), [118] | 0.25 (−0.70, 1.21) |
| Mean total WBC, ×10⁹/L (s.d.) | +2 | 13.8 (5.1), [128] | 13.0 (3.6), [121] | −0.84 (−2.03, 0.36) |
| | +6 | 13.5 (4.6), [121] | 12.9 (3.3), [118] | −0.58 (−1.79, 0.63) |
| Mean neutrophil count, ×10⁹/L (s.d.) | +2 | 4.5 (2.4), [128] | 4.4 (2.1), [121] | −0.06 (−0.66, 0.54) |
| | +6 | 4.9 (2.3), [120] | 4.5 (1.8), [118] | −0.30 (−0.90, 0.29) |
| Mean lymphocyte count, ×10⁹/L (s.d.) | +2 | 7.2 (3.0), [128] | 6.8 (2.4), [121] | −0.45 (−1.09, 0.19) |
| | +6 | 6.8 (2.9), [120] | 6.6 (2.5), [118] | −0.23 (-0.91, 0.45) |
| Mean monocyte count, ×10⁹/L (s.d.) | +2 | 1.1 (0.5), [128] | 1.0 (0.5), [121] | −0.06 (−0.20, 0.08) |
| | +6 | 1.0 (0.4), [120] | 0.9 (0.4), [117] | −0.03 (−0.17, 0.11) |
| Mean platelet count, ×10⁹/L (s.d.) | +2 | 486 (166), [128] | 496 (170), [121] | 6.74 (−35.59, 49.08) |
| | +6 | 483 (143), [121] | 501 (186), [118] | 15.53 (−28.27, 59.34) |
| Faecal inflammatory markers | | | | |
| Mean log₁₀ (faecal myeloperoxidase, ng/ml) (s.d.) | +2 | 7.79 (1.08), [116] | 7.65 (1.14), [115] | −0.18 (−0.51, 0.15) |
| | +6 | 7.38 (1.01), [113] | 7.42 (1.14), [109] | 0.03 (−0.31, 0.36) |
| Mean log₁₀ (faecal neopterin, nmol/L), (s.d.) | +2 | 6.45 (1.54), [116] | 6.65 (1.50), [115] | 0.20 (−0.16, 0.57) |
| | +6 | 6.37 (1.63), [114] | 6.04 (1.49), [109] | −0.36 (−0.72, -0.00) |
| Mean log₁₀(faecal alpha-1 antitrypsin, mg/dL) (s.d.) | +2 | 2.60 (1.11), [116] | 2.83 (0.78), [115] | 0.21 (−0.05, 0.47) |
| | +6 | 2.64 (0.87), [114] | 2.86 (0.83), [109] | 0.21 (−0.04, 0.47) |

*25(OH)D* 25-hydroxyvitamin D, *CI* confidence interval, *MDAT* Malawi Developmental Assessment Tool, *PTH* parathyroid hormone. *S.d.* standard deviation, *WBC* white blood cell.
aadjusted for site of recruitment and baseline value.
bprimary outcome, $P = 0.89$.

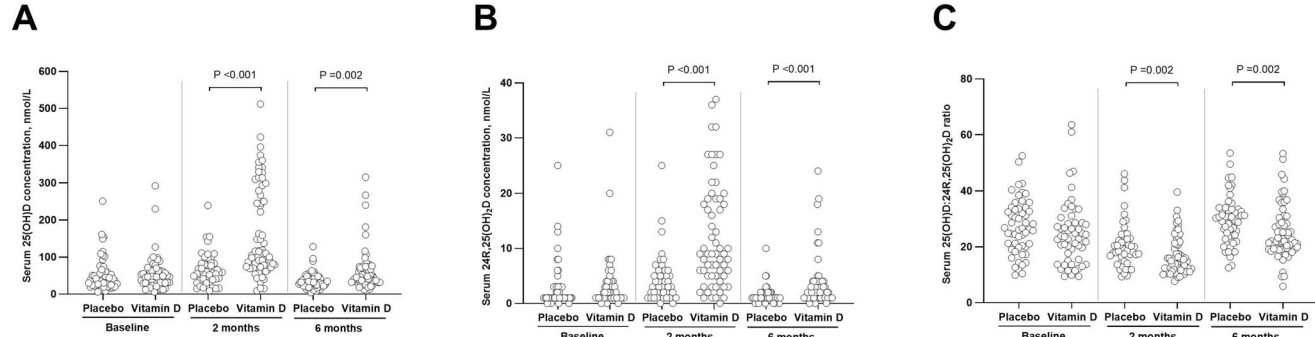

**Fig. 2 | Serum concentrations of vitamin D metabolites by allocation and timepoint. A** total 25(OH)D concentrations; (**B**) total 24 R,25(OH)₂D concentrations; (**C**) 25(OH)D-to-24R,25(OH)₂D ratios. *P* values are from generalised mixed models, testing for effect of allocation to vitamin D vs placebo with adjustment for baseline value and study site. Statistical tests were 2-sided, with no adjustment for multiple comparisons. Source data are provided as a Source Data file.

discharge and the second dose administered 14 days later, with a tolerance of 10–24 days.

**Standard of care**

All participants received standard treatment for complicated SAM as per Pakistan national guidelines[34], irrespective of allocation. These recommend administration of F-75 therapeutic milk (which contains 2.6–4.9 µg vitamin D₃ and 77 mg calcium per 100 mL) during the stabilisation phase, along with broad-spectrum antibiotics, folic acid, and any other specific treatment indicated for complications precipitating hospitalisation. Thereafter, F-100 therapeutic milk (containing 2.9–5.7 µg vitamin D₃ and 78 mg calcium per 100 mL) was given during a three-day transition phase, followed by administration of F-100 therapeutic milk or RUTF (which contains 15–20 µg vitamin D₃ and

300–600 mg calcium per sachet) during the rehabilitation phase. Once children were ready for discharge from the ward, they continued to receive RUTF as part of their outpatient therapeutic program, with weekly follow-up until nutritional rehabilitation was achieved. Iron was not routinely given during the stabilisation phase. If haemoglobin was less than 4 g/dl, then a 10 ml/kg blood transfusion was given with a diuretic in mid-transfusion. If the child was moderately anaemic and taking RUTF (which contains 10 mg iron per 100 g), the total iron requirement was calculated based on the child's weight (3–6 mg iron per kg body weight), and if this requirement was not being met by RUTF intake the necessary supplemental dose of iron was administered. A full blood count was repeated one month later, and results were used to guide a decision regarding the need for further iron supplementation thereafter.

**Table 3 | Safety outcomes by allocation**

| | | Placebo (n = 129)[a] | | Vitamin D (n = 127)[b] | |
|---|---|---|---|---|---|
| | | No. of events | Proportion of participants with one or more events (%) | No. of events | Proportion of participants with one or more events (%) |
| Fatal serious adverse event[c] | | 2 | 2/128 (1.6) | 0 | 0/123 (0.0) |
| Non-fatal serious adverse event[c] | | 9 | 9/128 (7.0) | 3 | 3/123 (2.4) |
| Non-serious adverse event leading to discontinuation of study supplement | | 0 | 0/128 (0.0) | 0 | 0/123 (0.0) |
| Relapse of SAM (complicated or uncomplicated) | | 4 | 4/128 (3.1) | 3 | 3/123 (2.4) |
| Other monitored safety conditions | Hypercalcaemia[d] | 6 | 5/128 (3.9) | 2 | 2/123 (1.6) |
| | Hypervitaminosis D[e] | 1 | 1/83 (1.2) | 22 | 22/85 (25.9) |
| | Hypercalciuria[f] | 56 | 50/128 (39.1) | 51 | 44/123 (35.8) |

[a]no. of participants who took at least one dose of placebo and for whom follow-up data were available for at least one timepoint.
[b]no. of participants who took at least one dose of vitamin D and for whom follow-up data were available for at least one timepoint.
[c]causes for each event presented in Table S5, Supplementary Information.
[d]defined as serum albumin-adjusted calcium concentration >2.65 mmol/L.
[e]defined as 25(OH)D concentration >220 nmol/L.
[e]defined as urinary calcium:creatinine molar ratio >1.00.

## Outcomes

The primary outcome was the difference between active and placebo participants in mean weight-for-height or -length z-score (WHZ) at 2 months after administration of the first dose of IMP, calculated using WHO child growth standards[35] and accounting for baseline values. Secondary efficacy outcomes were inter-arm differences in mean WHZ at 6-month follow-up; mean weight-for-age z-score, height- or length-for-age z-score, head circumference-for-age z-score, mid-upper arm circumference and lean mass index (calculated as 1000/impedance at 50 kHz) at 2- and 6-month follow-up; mean MDAT scores (overall and gross motor, fine motor, language and social scores separately) at 2- and 6-month follow-up; serum concentrations of albumin, C-reactive protein, total 25-hydroxyvitamin D (25[OH]D, the accepted biomarker of vitamin D status), total 24R,25-dihydroxyvitamin D (24R,25[OH]₂D, the major catabolite of 25[OH]D), ratios of serum concentrations of total 25(OH)D to total 24R,25[OH]₂D (indicating the degree of interconversion of these metabolites), total alkaline phosphatase, parathyroid hormone, ferritin and hepcidin at 2- and 6-month follow-up; mean urinary molar ratios of calcium:creatinine and mean urine osmolality at 2- and 6-month follow-up; full blood count indices (mean haemoglobin concentration, mean corpuscular volume, mean corpuscular haemoglobin concentration, mean total and differential white blood cell counts) at 2- and 6-month follow-up; and median faecal concentrations of inflammatory markers (myeloperoxidase, neopterin and alpha-1 antitrypsin) at 2- and 6-month follow-up. Safety outcomes were mortality, incidence of serious adverse events, the proportion of participants experiencing relapse of SAM (complicated or uncomplicated) over 6-month follow-up, and incidence of hypercalcaemia (defined as serum albumin-adjusted calcium >2.65 mmol/L), hypercalciuria (defined as urinary molar calcium: creatinine ratio >1.00), hypervitaminosis D (defined as serum 25[OH]D concentration >220 nmol/L) or any other adverse event attributed to IMP.

## Laboratory assays

Serum analysis of 25(OH)D, 24R,25[OH]₂D, albumin, C-reactive protein (CRP), alkaline phosphatase (ALP), intact parathyroid hormone (PTH), ferritin and hepcidin were performed at the Bioanalytical Facility, University of East Anglia (Norwich, UK) and undertaken in Good Clinical and Laboratory Practice conditions. 25(OH)D₃, 25(OH)D₂, 24R,25[OH]₂D₃ and 24R,25[OH]₂D₂ were measured separately using liquid chromatography-tandem mass spectrometry (LC-MS/MS) as

previously described[36] with values of D₂ vs D₃ metabolites summed to yield values for total concentration. The assays were calibrated using standard reference material SRM972a from the National Institute of Science and Technology, and the assay showed linearity between 0 and 200 nmol/l. The inter/intra-assay coefficient of variation (CV) across the assay range was ≤9% and the lower limit of quantification was 0.1 nmol/L. The assay showed an accuracy bias of −2.9 to +5.7% against the vitamin D external quality assessment scheme (DEQAS) LC-MS/MS Reference Measurement Procedure provided by the Centers for Disease Control and Prevention (CDC, Atlanta, GA, USA) (http://www.deqas.org/; accessed on April 15, 2024). Serum concentrations of albumin, CRP, ALP, PTH, and ferritin were analysed using the COBAS c501 and e601 automated platforms (Roche Diagnostics, Mannheim, Germany) according to the manufacturer's instructions. The inter-assay CVs and assay range (%CV, range) for albumin were (≤3%, 2.0–100 g/L), CRP (≤4.2%, 0.6−350 mg/L), ALP (≤3%, 5−1200 U/L), PTH (≤3.8%, 0.127-530 pmol/L) and ferritin (≤5.5%, 0.5−2000 μg/L). Serum hepcidin was analysed by the Simple Plex Human Hepcidin assay kit (PN# SPCKB-PS-000984) using the Ella automated immunoassay platform (Bio-Techne, Minneapolis, MN, USA) according to the manufacturer's instructions. The inter-assay CVs were ≤8.2% between the assay range 9.7 and 37,000 pg/mL. Serum concentrations of calcium and albumin were measured at Chughtai Laboratory (Lahore, Pakistan) by photometric methods on the Abbott Alinity ci (Abbott Diagnostics, UK) according to the manufacturer's instructions. The inter-assay CVs for total calcium and albumin were ≤2.1% across the assay working ranges of 0.2–7.5 mmol/l and 2−60 g/l. Albumin-adjusted calcium (mmol/L) was calculated as measured total calcium (mmol/L) + 0.02 × (40 − serum albumin [g/L]). Urine concentrations of calcium and creatinine and urine osmolarity were measured at Chughtai Laboratory (Lahore, Pakistan) by photometric methods & Freezing Point Depression on the Abbott Alinity ci & OSMO-1 platforms (Abbott Diagnostics, UK and Advanced Instruments, US, respectively) according to the manufacturer's instructions. The inter-assay CVs for urinary calcium, urinary creatinine and urinary osmolarity were 0.9%, 0.5% and 0.5%, respectively. Urinary calcium:creatinine ratios were calculated using molar concentrations of each analyte. Full blood counts were analysed at Chughtai Laboratory (Lahore, Pakistan) on the Sysmex XN-1000 platform (Sysmex, Japan) according to the manufacturer's instructions.

## Sample size

Assuming a mean 2-month WHZ of −2.0 in the control arm[37] with a standard deviation of 1.25 and 20% loss to follow-up, we calculated that

a total of 250 participants (125 per arm) would allow us to detect an inter-arm difference in WHZ of 0.50 (−1.5 vs. −2.0 in intervention vs control arms, respectively) with 80% power and 5% type 1 error.

## Statistical methods

Statistical analyses were performed using Stata software (Version 17.0; StataCorp, College Station, Texas, United States) and SAS (Version 9.4; SAS Institute, Cary NC) according to the Statistical Analysis Plan available at https://osf.io/7h9cu?view_only=e9675bfa22c747199d811eb8a50be10d. Distributions of outcome variables were assessed and where non-normal, data were log-transformed. All follow-up efficacy outcomes accounted for site of recruitment and their baseline value. There was no hierarchical structure for secondary and safety outcomes and there was no adjustment for multiplicity for these outcomes. All statistical analyses were performed on an intention-to-treat basis and conducted two-sided. $P$ values < 0.05 were interpreted as significant. The effect of allocation on the primary outcome of mean WHZ at 2-month follow-up was analysed using generalised mixed models, including baseline and endpoint values for each subject, linked with random intercept terms. The models tested for effect of allocation to vitamin D vs placebo with adjustment for baseline value and study site (which was a stratification factor). The effect of allocation on secondary efficacy outcomes was analysed as follows: continuous outcomes (such as mean anthropometric z-scores, neuro-developmental scores, lean mass index) were analysed using analogous models to the primary outcome. Dichotomous efficacy outcomes (such as the proportion of participants experiencing a given outcome at least once) were analysed using non-linear mixed models with a logit link and binomial/mixed error, analogous to the primary analysis. The effect of allocation on safety outcomes was investigated in all participants who took one or more doses of IMP, and is described narratively. Heterogeneity of treatment effect was examined among sub-groups defined by baseline vitamin D status (<50 nmol/L vs. ≥50 nmol/L), baseline WHZ (< −3.00 vs. ≥−3.00) and sex (male vs. female), as pre-defined in the Statistical Analysis Plan (available at https://osf.io/7h9cu?view_only=e9675bfa22c747199d811eb8a50be10d). For these analyses, the primary analysis was repeated with additional inclusion of an interaction term between the sub-group and allocation to vitamin D vs. placebo; a P value for the interaction between allocation and sub-group is presented.

## Inclusion and ethics

The research included local researchers throughout the research process—study design, study implementation, data ownership, intellectual property and co-authorship of this publication. The research is locally relevant, and this has this been determined in collaboration with local partners. Roles and responsibilities were agreed amongst collaborators ahead of the research and capacity-building plans for local researchers were discussed. The study has been approved by ethics committees in Pakistan. The research did not result in stigmatisation, incrimination, discrimination or otherwise personal risk to participants. The research did not involve health, safety, security or other risk to researchers. Benefit sharing measures have been discussed in relation to biological materials transferred out of the country. We have taken local and regional research relevant to our study into account in citations.

## Reporting summary

Further information on research design is available in the Nature Portfolio Reporting Summary linked to this article.

## Data availability

Individual de-identified data generated in this study and an accompanying data dictionary are immediately and indefinitely available in a codeocean database, under accession code 2673880. These data can be used for any purpose. Source data and a statistical analysis plan are provided with this paper. Source data are provided with this paper.

## Code availability

The code generated in this study has been deposited in a codeocean database under accession code 2673880.

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

## Acknowledgements

We thank all the children who participated in the trial, and their parents and guardians. We also thank members of the Independent Data Monitoring Committee (Prof James Berkley [Chair], University of Oxford, Oxford, UK; Prof Munir Saleemi, The University of Lahore, Lahore, Pakistan; and Dr Charles Opondo [Statistician], London School of Hygiene and Tropical Medicine, London, UK) and the Trial Steering Committee (Prof Suzanne Filteau [Chair], London School of Hygiene and Tropical Medicine, London, UK; Prof Nosheen Javed, University of Veterinary and Animal Sciences (UVAS), Lahore, Pakistan; and Dr Kelsey Jones, Great Ormond St Hospital, London, UK). Trial monitoring was performed by DRK Pharma Solutions, Lahore, Pakistan. The trial was funded by the Thrasher Research Fund (ref 15080), who had no role in study design, implementation, analysis or data interpretation.

## Author contributions

J.S., R.Z., A.J.P., and A.R.M. conceived the study and contributed to study design and protocol development. J.C. contributed to design of, and training in, developmental assessments. J.P. contributed to design of, and training in, body composition assessments. J.S. led on trial implementation, with support from R.Z., M.S.B., R.K., A.C., and A.R.M. R.K. and A.C. acted as lead clinical investigators at study sites. J.C.Y.T. and W.D.F. supervised measurements of serum concentrations of vitamin D metabolites, and Z.A. supervised measurements of faecal concentrations of inflammatory markers. D.A.J., N.F., and A.R.M. wrote the statistical analysis plan. J.S., M.S.B., and D.A.J. managed data. J.S., D.A.J., and A.R.M. accessed and verified the data underlying the study. D.A.J. and N.F. conducted statistical analyses. J.S., D.A.J., and A.R.M. wrote the first draft of the trial report. All other authors made substantive comments thereon and approved the final version for submission. J.S. and A.R.M. are the guarantors.

## Competing interests

ARM declares receipt of funding in the last 36 months to support vitamin D research from the following companies who manufacture or sell vitamin D supplements: Pharma Nord Ltd, DSM Nutritional Products Ltd, Thornton & Ross Ltd and Hyphens Pharma Ltd. ARM also declares receipt of vitamin D capsules for clinical trial use from Pharma Nord Ltd, Synergy Biologics Ltd and Cytoplan Ltd; support for attending meetings from Pharma Nord Ltd and Abiogen Pharma Ltd; receipt of consultancy fees from DSM Nutritional Products Ltd and Qiagen Ltd; receipt of a speaker fee from the Linus Pauling Institute; participation on Data and Safety Monitoring Boards for the VITALITY trial (Vitamin D for Adolescents with HIV to reduce musculoskeletal morbidity and immunopathology, Pan African Clinical Trials Registry ref PACTR20200989766029) and the Trial of Vitamin D and Zinc Supplementation for Improving Treatment Outcomes Among COVID-19 Patients in India (ClinicalTrials.gov ref NCT04641195); and unpaid work as a Programme Committee member for the Vitamin D Workshop. All other authors declare that they have no competing interests.

## Additional information

Javeria Saleem ⬥[1] ✉, Rubeena Zakar[1], Muhammad Salman Butt[1,2], Rameeza Kaleem[3], Asif Chaudhary[4], Jaya Chandna[5],
David A. Jolliffe[6], Joseph Piper[6], Zaigham Abbas[7], Jonathan C. Y. Tang ⬥[8,9], William D. Fraser[8,9], Nick Freemantle[10],
Andrew J. Prendergast ⬥[6] & Adrian R. Martineau ⬥[6] ✉

[1]Department of Public Health, University of the Punjab, Lahore, Punjab, Pakistan. [2]Institute of Nursing and Health Research, Ulster University, Belfast, UK. [3]Sir
Ganga Ram Hospital, Lahore, Punjab, Pakistan. [4]THQ Hospital, Lahore, Punjab, Pakistan. [5]Department of Infectious Disease Epidemiology and International
Health, Faculty of Epidemiology and Population Health, London School of Hygiene and Tropical Medicine, London, UK. [6]Blizard Institute, Faculty of Medicine
and Dentistry, Queen Mary University of London, London, UK. [7]Institute of Microbiology and Molecular Genetics, University of the Punjab, Lahore,
Punjab, Pakistan. [8]Norwich Medical School, University of East Anglia, Norwich, UK. [9]Departments of Laboratory Medicine, Clinical Biochemistry and
Departments of Diabetes and Endocrinology, Norfolk and Norwich University Hospital NHS Foundation Trust, Norwich, UK. [10]Comprehensive Clinical Trials
Unit, University College London, London, UK. ✉e-mail: javeria.hasan@hotmail.com; a.martineau@qmul.ac.uk

