## [Transparent Peer Review file · Nature Communications]

High-dose vitamin D3 to improve outcomes in the convalescent phase of complicated severe acute malnutrition in Pakistan: a double-blind randomised controlled trial (ViDiSAM)

Corresponding Author: Professor Adrian Martineau

Version 0:

Reviewer comments:

Reviewer #1

(Remarks to the Author)

This is an interesting trial of high dose vitamin D (two doses of 200,000 units each) among children with complicated severe acute malnutrition (SAM) admitted to two rehabilitation facilities in Pakistan. The trial follows a previous study by the same group of authors (Saleem et al AJCN 2018) among children with uncomplicated SAM with demonstrated benefits on weight for age showing some benefits in weight gain and developmental outcomes at 8 weeks post treatment.

The trial appears to have been done with reasonable fidelity as a double-blind study (the protocol says quadruple blind) and as an essentially negative study, is an important addition to the literature. There are several issues with the execution and interpretation of the data that do merit clarification and reconsideration.

1. The design:

- a. Given the importance and putative benefits of vitamin D in SAM, why was this not administered at admission or an earlier stage of treatment? Giving the first dose at discharge or just prior to it, essentially makes this a study in recovered or recovering children with SAM, and presumably after complications settled.
- b. We are not provided any information of other treatments (especially nutritional therapy including micronutrients) that these children might have received prior to and during hospitalization. Description of the standard of care in this population (other than RUTF) would help understand what else was done. Was anything done for these universally anemic children for example?
- c. The use of the Malawi developmental score in this population versus other developmental assessment tools already validated in various Pakistani children, needs further justification. Why was this chosen in preference to other tools including locally adapted Bayley's development tests?

2. It is unclear to this reviewer as to why serum Vit D concentrations at baseline and end-line are only available in 47 and 62 children among placebo and Vit D groups respectively, when all 273 children provided baseline blood samples at recruitment? The post-hoc findings of better outcomes among a subgroup of children seems forced as this analysis was neither pre-specified in the protocol, nor does the cut off of 50 nmol/l make sense given mean baseline values which were not too dissimilar (51.8 and 57.1 nmol/l respectively). This subgroup analysis should be dropped for a number of reasons including lack of information on representativeness.

3. There is ample information in Pakistan on vitamin D status at population level for women of reproductive age and children (National Nutrition Survey 2018). Was any effort made to look at maternal status or indeed the comparison of this populations' status with such surveys? The prevalence of deficiency in 27.8% and 18.9% of the children at recruitment is much lower than population figures and merits some explanation, apart from the imbalance between the two groups at admission.

(Remarks on code availability)

Reviewer #3

(Remarks to the Author)

Excellent manuscript, I have some suggested improvements:

- give latitude of the center
- clarify no of subjects in the subgroup analyses
- can you explain the high no. of hypercalcemia (n=7) in the excluded patients?
- it is unclear to me if standard rickets prophylaxis was given ?

(Remarks on code availability)

Reviewer #4

(Remarks to the Author)

The authors have conducted a placebo-controlled trial to evaluate the effect of high-dose vitamin D3 on weight gain and neurodevelopmental indices in children with severe acute malnutrition (SAM). Although the authors observed higher mean serum 25(OH)D concentrations in children receiving high-dose vitamin D3 compared to children in the placebo group, there were no differences in any outcome measures between groups. The study is interesting but some additional clarifications/modifications are required to validate the findings obtained in the study:

1. High-risk deliveries including pre-term or some known maternal characteristics (pre-eclampsia or gestational diabetes) are associated with low birth weight and need to be accounted for in the analyses. If you remove the cases with high-risk deliveries including pre-term birth and low-birth weights whether the differences in intervention groups or primary outcomes are still the same or different.
2. Please provide the references for including the input values in the sample size computation.
3. In Table 1, there is no reason to report the distribution of baseline characteristics for the overall patients. I would suggest removing this column and reporting the p-value to ensure the randomization is achieved or not.
4. In Table 2, please highlight significant p-values.
5. P-values were not reported for fecal inflammatory markers. Also, please do not compute the p-value using a nonparametric test on log-transformed data. Either use a nonparametric test on the original data or use a parametric test on the log-transformed data.
6. Table 3, extremely unbalanced sample sizes were observed between baseline 25(OH)D ≥ 50 vs. < 50 , and therefore a Welch t-test or nonparametric tests should be preferred without using the random effects models.
7. Mean serum 25(OH)D was only available for 1/2 to 1/3 of the patients in each group. However, safety outcomes of hypervitaminosis D were observed for all participants.
8. Due to separate groups in serum 25(OH)D concentrations at 2 months in the active group, it would be worth determining a Spearman rank correlation between serum 25(OH)D concentrations and primary outcomes within each treatment group.
9. Please follow CONSORT reporting guidelines.

(Remarks on code availability)

Version 1:

Reviewer comments:

Reviewer #1

(Remarks to the Author)

The authors have provided reasonable responses to the review of their trial but I am afraid, skirted some of the questions and implications therein

1. The point about timing of vitamin D administration was answered in the context of "mirroring" their previous trial. WHO recommendations for the treatment of severe acute malnutrition do NOT preclude administration of needed micronutrients during in-hospital treatment. In fact it is expected standard of care. If the objective here was to improve outcomes in the convalescent phase of treatment of such children, it should be clearly reflected in the title and trial description.
2. The issue with the laboratory measurement challenges (mislabeling) now explained, does raise questions on trial quality and should be mentioned clearly in the trial description. I appreciate the information provided on the subset studied and would suggest that this information be included in the paper. Moving the analysis of outcomes by vit D status to supplementary material is prudent.
3. Comment 3 also pertained to vit D status in children available from the National Nutrition Survey 201, and comparison to the values found in the trial among severely malnourished children. Have the authors considered this point in terms of external validity of their findings and subjects?

(Remarks on code availability)

I have not attempted to run the code or analyse the data myself

Reviewer #3

(Remarks to the Author)

I am happy with the extensive revisions made that addressed all the reviewers' comments.

(Remarks on code availability)

Reviewer #4

(Remarks to the Author)

The authors have thoroughly addressed all my concerns and I do not have any further concerns.

(Remarks on code availability)

High-dose vitamin D3 in the treatment of complicated severe acute malnutrition in Pakistan: a double-blind randomised controlled trial (ViDiSAM): NCOMMS-24-44517.

Response to reviewer comments

Reviewer #1, Comment #1a: *Given the importance and putative benefits of vitamin D in SAM, why was this not administered at admission or an earlier stage of treatment? Giving the first dose at discharge or just prior to it, essentially makes this a study in recovered or recovering children with SAM, and presumably after complications settled.*

Response: Critically unwell children who are clinically unstable are not an easy population to intervene in, and discernment of serious adverse events in this group is also challenging. We know that the pathology of SAM persists beyond the early days of admission, having demonstrated persistent inflammation out to 48-week follow-up (1) associated with high outpatient mortality and readmission, so there is increasing interest in developing interventions to improve outcomes in the post-discharge window. This is underlined by the latest WHO guidelines for the management of SAM (2), which highlight the evidence gap in improving convalescent care.

We have added the following text to the Introduction to clarify the basis for intervening just before hospital discharge:

‘We chose to mirror the protocol of our previous trial by intervening at the point of hospital discharge rather than at the point of hospital admission, in order to ameliorate pathology that persists beyond the period of hospitalisation (1) and to address the evidence gap in improving convalescent care, highlighted by the latest WHO Guidelines for the management of SAM (2).’

Reviewer #1, Comment #1b. *We are not provided any information of other treatments (especially nutritional therapy including micronutrients) that these children might have received prior to and during hospitalization. Description of the standard of care in this population (other than RUTF) would help understand what else was done. Was anything done for these universally anemic children for example?*

Response: We have added the following section to the Methods section of the revised manuscript to clarify details of standard of care:

‘All participants received standard treatment for complicated SAM as per Pakistan national guidelines (3), irrespective of allocation. These recommend administration of F-75 therapeutic milk (which contains 2.6-4.9 µg vitamin D₃ and 77 mg calcium per 100 mL) during the stabilisation phase, along with broad-spectrum antibiotics, folic acid, and any other specific treatment indicated for complications precipitating hospitalisation. Thereafter, F-100 therapeutic milk (containing 2.9-5.7 µg vitamin D₃ and 78 mg calcium per 100 mL) was given during a three-day transition phase, followed by administration of F-100 therapeutic milk or RUTF (which contains 15-20 µg vitamin D₃ and 300-600 mg calcium per sachet) during the rehabilitation phase. Once children were ready for discharge from the ward, they continued to receive RUTF as part of their outpatient therapeutic program, with weekly follow-up until nutritional rehabilitation was achieved. Iron was not routinely given during the stabilisation phase. If haemoglobin was less than 4 g/dL, then a 10 mL/kg blood transfusion was given with a diuretic in mid-transfusion. If the

child was moderately anaemic and taking RUTF (which contains 10 mg iron per 100 g), the total iron requirement was calculated based on the child's weight (3-6 mg iron per kg body weight), and if this requirement was not being met by RUTF intake the necessary supplemental dose of iron was administered. A full blood count was repeated one month later, and results were used to guide a decision regarding the need for further iron supplementation thereafter.'

Reviewer #1, Comment #1c. *The use of the Malawi developmental score in this population versus other developmental assessment tools already validated in various Pakistani children, needs further justification. Why was this chosen in preference to other tools including locally adapted Bayley's development tests?*

Response: We agree that we could have chosen multiple different tools, and highlight that there is no consensus on the "best" tools to use in different settings. The Malawi Development Assessment Tool, despite the name, has been used across multiple settings, including Pakistan (4), and works well across the age range of children in this study, as a directly observed assessment. Moreover, the test yields a continuous read-out, across multiple domains, that is more sensitive for detection of an effect of the intervention than DDST that we used in our previous trial (5).

We have added the following text to the Introduction of the revised manuscript to clarify the reasons for selection of MDAT in this trial:

'We used the Malawi Development Assessment Tool (MDAT) to assess neurodevelopmental status, as this instrument has been used across multiple settings, including Pakistan (4), and works well across the age-range of children eligible to participate in this study as a directly observed assessment. Moreover, the MDAT yields a continuous read-out, across multiple domains, that is more sensitive for detection of an effect of the intervention than the Denver Developmental Screening Test that we used in our previous trial (5).'

Reviewer #1, Comment #2. *It is unclear to this reviewer as to why serum Vit D concentrations at baseline and end-line are only available in 47 and 62 children among placebo and Vit D groups respectively, when all 273 children provided baseline blood samples at recruitment? The post-hoc findings of better outcomes among a subgroup of children seems forced as this analysis was neither pre-specified in the protocol, nor does the cut off of 50 nmol/l make sense given mean baseline values which were not too dissimilar (51.8 and 57.1 nmol/l respectively). This subgroup analysis should be dropped for a number of reasons including lack of information on representativeness.*

Response: We acknowledge the limitations of this subgroup analysis. However, it was pre-specified both in the study protocol and in the statistical analysis plan, so GCP obliges us to report its findings. We have therefore de-emphasised it by excising references to it from the abstract, and moving the relevant Table from the main manuscript to supplementary material (new Table S5).

To address specific queries raised by the reviewer: the 50 nmol/L threshold used for this sub-group analysis was selected on the basis of widespread acceptance of this value as a cut-off to define vitamin D deficiency, as recommended by the US Institute of Medicine.⁽⁶⁾ Similarity of baseline serum 25(OH)D concentrations for participants allocated to vitamin D vs. placebo is expected, given that they were randomised. The fact that mean baseline values were around 50 nmol/L does not invalidate a sub-group analysis that classifies participants as deficient or sufficient using the 50 nmol/L threshold.

Serum 25(OH)D concentrations are missing for some children due to problems with sample labelling at Chughtai laboratory. We have highlighted this as a limitation in the Discussion ('Finally, serum samples were only available for biochemical analysis for a subset of participants, which limited power for sub-group analyses by baseline vitamin D status.')

We highlight that quality control checks conducted during preparation of this revision revealed an error in the code used to generate results of this sub-group analysis, which has now been corrected. P values for interaction are all >0.05 (Table S5).

With regard to characteristics of those for whom baseline serum 25(OH)D concentrations are available: please see Rebuttal Table 1 below, which confirms that they are broadly representative of the study population as a whole.

Rebuttal Table 1. Baseline participant characteristics, overall vs. those in sub-group analysis by baseline vitamin D status

		Overall (n=259)	Participants included in subgroup analysis by baseline serum 25(OH)D concentration (n=106)
Sociodemographic			
Mean age, months (s.d.) [range]		14.5 (8.2) [6.0-49.7]	14.0 (7.0) [6.0 to 39.9]
Sex, n (%)	Female	147 (56.8%)	60 (56.6%)
	Male	112 (43.2%)	46 (43.4%)
Ethnic origin, n (%)	Kashmiri	1 (0.4%)	-
	Muhajir	5 (1.9%)	-
	Pashtoon	10 (3.9%)	4 (3.8%)
	Punjabi	211 (81.5%)	93 (87.7%)
	Saraiki	2 (0.8%)	-
	Other	30 (11.6%)	9 (8.5%)
Study site, n (%)	Sir Ganga Ram Hospital	129 (49.8%)	59 (55.7%)
	THQ Hospital	130 (50.2%)	47 (44.3%)
Number of younger siblings at home, n (%)	0	237 (91.9%)	97 (92.4%)
	1	18 (7.0%)	7 (6.7%)
	2 or more	3 (1.2%)	1 (1.0%)
Number of older siblings at home, n (%)	0	42 (16.2%)	14 (13.2%)
	1	74 (28.6%)	38 (35.8%)
	2	57 (22.0%)	22 (20.8%)
	3	37 (14.3%)	9 (8.5%)
	4	31 (12.0%)	14 (13.2%)
	5 or more	18 (6.9%)	9 (8.5%)
Mother's education, n (%)	No formal schooling	92 (35.5%)	44 (41.5%)
	Primary education only	91 (35.1%)	30 (28.3%)
	Secondary education and above	76 (29.3%)	32 (30.2%)
Father's education, n (%)	No formal schooling	96 (37.1%)	46 (43.4%)
	Primary education only	85 (32.8%)	35 (33.0%)
	Secondary education and above	78 (30.1%)	25 (23.6%)

Mean monthly household income, PKR (s.d.)		22575 (14110)	20264 (7576)
Type of residence	Cottage	1 (0.4%)	-
	Flat/apartment	227 (87.6%)	92 (86.8%)
	Terraced house	22 (8.5%)	12 (11.3%)
	Other	9 (3.5%)	2 (1.9%)
Vaccination status, n (%)	Up-to-date	178 (68.7%)	68 (64.2%)
	Not up-to-date	81 (31.3%)	38 (35.8%)
Early life characteristics			
Gestational age, n(%)	Term	233 (90.0%)	95 (89.6%)
	Pre-term	26 (10.0%)	11 (10.4%)
Child's birthweight, n (%)	Very low birth weight (<1.5 kg)	2 (0.8%)	-
	Low birth weight (1.5 to < 2.5 kg)	108 (41.7%)	37 (34.9%)
	Normal (≥2.5 kg < 4.0 kg)	146 (56.4%)	67 (63.2%)
	Too heavy (≥ 4.0 kg)	3 (1.2%)	2 (1.9%)
Breastfeeding, n (%)	Still breastfeeding	92 (35.5%)	31 (29.2%)
	Breastfed for 0-5 months then	115 (44.4%)	51 (48.1%)
	Breastfed for 6-12 months then	22 (8.5%)	13 (12.3%)
	Breastfed >12 months then	30 (11.6%)	11 (10.4%)
Age at weaning, months, n (%)	Still exclusively breastfeeding	3 (1.2%)	1 (0.9%)
	<4 months	8 (3.1%)	4 (3.8%)
	4-5 months	67 (25.9%)	27 (25.5%)
	6-8 months	151 (58.3%)	60 (56.6%)
	9-11 months	21 (8.1%)	8 (7.5%)
	12-23 months	9 (3.5%)	6 (5.7%)
Anthropometry at baseline			
Mean Weight-for-height or -length z-		-3.5 (0.9)	-3.6 (0.9)
Mean mid-upper arm circumference, cm		10.5 (0.9)	10.5 (0.9)
Bilateral pitting oedema at hospital admission, n (%)	Present	0 (0%)	0 (0%)
	Absent	259 (100.0%)	106 (100.0%)
Mean weight-for-age z-score (s.d.)		-4.2 (1.0)	-4.2 (1.0)
Mean head circumference z-score (s.d.)		-2.8 (1.2)	-3.9 (0.9)
Lean mass index, Ω^{-1} (s.d.)		11.9 (2.0)	11.9 (2.0)
Complication present at hospital admission, n (%)	Acute lower respiratory infection	51 (19.7%)	14 (13.2%)
	Acute upper respiratory infection	19 (7.3%)	7 (6.6%)
	Anorexia	8 (3.1%)	3 (2.8%)
	Gastroenteritis / Diarrhoea	104 (40.2%)	56 (52.8%)
	Hyperpyrexia	6 (2.3%)	2 (1.9%)
	Hypoglycaemia	1 (0.4%)	-
	Severe anaemia	16 (6.2%)	3 (2.8%)
	Severe dehydration	44 (17.0%)	16 (15.1%)
	Other	10 (3.9%)	5 (4.7%)
Neurodevelopmental status at baseline			
Mean overall MDAT score (all domains)		50.8 (21.0)	48.6 (19.9)
Mean Gross motor score (s.d.)		13.6 (5.3)	13.0 (5.3)
Mean Fine motor score (s.d.)		13.9 (5.8)	13.2 (5.4)
Mean Language score (s.d.)		10.4 (5.5)	9.8 (4.9)
Mean Social score (s.d.)		12.9 (5.4)	12.6 (5.3)
Blood biochemistry			

Mean total serum 25(OH)D, nmol/L (s.d.)		54.4 (47.3)	54.1 (47.3)
Total serum 25(OH)D category	<25 nmol/L, n (%)	25 (23.4%)	24 (22.6%)
	≥ 25 nmol/L & < 50 nmol/L, n (%)	43 (40.2%)	43 (40.6%)
	≥ 50 nmol/L & < 75 nmol/L, n (%)	19 (17.8%)	19 (17.9%)
	≥75 nmol/L, n (%)	20 (18.7%)	20 (18.9%)
Mean total serum 24R,25(OH) ₂ D, nmol/L		3.1 (4.7)	3.1 (4.7)
Mean total 25(OH)D: total 24R,25(OH) ₂ D		26.1 (10.8)	26.1 (10.8)
Mean serum albumin, g/L (s.d.)		37.7 (5.5)	37.4 (6.1)
Mean serum C-reactive protein, mg/L		5.1 (10.3)	5.2 (10.4)
Mean serum albumin-adjusted calcium, mmol/L (s.d.)		2.38 (0.18)	2.40 (0.20)
Mean serum total alkaline phosphatase,		278.5 (264.9)	275.4 (265.0)
Mean serum PTH, pmol/L (s.d.)		3.3 (5.7)	3.4 (5.9)
Mean serum ferritin, µg/L (s.d.)		53.8 (83.5)	54.6 (84.1)
Mean serum hepcidin, ng/ml (s.d.)		20.2 (39.1)	20.6 (39.4)
Urine biochemistry			
Mean urinary calcium creatinine molar		1.81 (5.75)	1.62 (4.63)
Mean urinary osmolality, mOsm/kg (s.d.)		641.03 (290.42)	593.1 (293.4)
Proportion with urinary		71/259 (27.4%)	32/106 (30.2%)
calcium creatinine molar ratio >1.0			
Mean haemoglobin, g/dL (s.d.)		9.0 (1.9)	9.2 (2.0)
Mean corpuscular volume, fL (s.d.)		69.8 (12.9)	70.0 (13.2)
Mean corpuscular haemoglobin		21.3 (4.9)	21.4 (5.1)
Mean total WBC, x10 ⁹ /L (s.d.)		12.5 (5.7)	12.8 (6.2)
Mean neutrophil count, x10 ⁹ /L (s.d.)		3.9 (2.7)	3.8 (2.2)
Mean lymphocyte count, x10 ⁹ /L (s.d.)		6.9 (3.2)	6.9 (3.2)
Mean monocyte count, x10 ⁹ /L (s.d.)		1.0 (0.7)	1.1 (0.8)
Mean platelet count, x10 ⁹ /L (s.d.)		455 (192)	439 (163)
Faecal inflammatory markers			
Mean log-transformed faecal myeloperoxidase, ng/ml (s.d.)		7.5 (1.4)	7.6 (1.5)
Mean log-transformed faecal neopterin, nmol/L (s.d.)		6.9 (1.3)	6.8 (1.4)
Mean log-transformed faecal alpha-1 antitrypsin, mg/dL (s.d.)		2.4 (1.1)	2.5 (1.0)

Abbreviations: THQ Hospital: Tehsil Headquarter Hospital, PKR: Pakistani Rupee, KHz: Kilohertz, MDAT: Malawi Developmental Assessment Tool, WBC: White blood cell, PTH: Parathyroid hormone, 25(OH)D: 25-hydroxyvitamin D, S.d.: Standard deviation, CI: Confidence interval.

Reviewer #1, Comment #3. *There is ample information in Pakistan on vitamin D status at population level for women of reproductive age and children (National Nutrition Survey 2018). Was any effort made to look at maternal status or indeed the comparison of this populations' status with such surveys? The prevalence of deficiency in 27.8% and 18.9% of the children at recruitment is much lower than population figures and merits some explanation, apart from the imbalance between the two groups at admission.*

Response: Maternal vitamin D status was not assessed in this study, as the focus was on evaluating effects of high-dose vitamin D in children by conducting an RCT. A survey of maternal

vitamin D status would indeed be interesting, but this would have been beyond the scope of a clinical trial.

Reviewer #3, Comment #1: *give latitude of the center*

Response: the latitude of Lahore is 31.5° N – Methods have been updated in revised manuscript to include this information.

Reviewer #3, Comment #2: *clarify no of subjects in the subgroup analyses*

Response: numbers contributing data to sub-group analyses are presented in square brackets, Table 3 (main manuscript) and Tables S3 and S4 (supplementary material).

Reviewer #3, Comment #3: *can you explain the high no. of hypercalcemia (n=7) in the excluded patients?*

Response: We are unable to confidently explain this. High-dose vitamin D is readily available in Pakistan, so it is possible that some children received this treatment prior to hospitalisation.

Reviewer #3, Comment #4: *it is unclear to me if standard rickets prophylaxis was given ?*

Response: Children with clinical signs of rickets at baseline were excluded from the trial and referred for further evaluation and treatment, as per text of Methods:

‘A physical examination was also performed to assess children for signs of rickets (genu varum, genu valgum, windswept deformity of the knees, rickety rosary, frontal bossing or wrist-widening). Children exhibiting such signs were excluded from the trial and referred to the nearest appropriate clinic for investigation and treatment.’

Children without clinical signs of rickets at baseline did not receive specific rickets prophylaxis. However, they did receive standard treatment for complicated SAM as per WHO guidelines during hospitalisation, which contains both calcium and vitamin D, i.e. F-75 therapeutic milk (which contains 2.6-4.9 mcg vitamin D3 and 77 mg calcium per 100 mL) during the stabilization phase, then F-100 therapeutic milk (containing 2.9-5.7 mcg vitamin D3 and 78 mg calcium per 100 mL) during a three-day transition phase, then F-100 therapeutic milk or RUTF (which contains 15-20 mcg vitamin D3 and 300-600 mg calcium per sachet) during the rehabilitation phase. Once children were ready for discharge from the ward, they continued to receive RUTF as part of their outpatient therapeutic program, with weekly follow-up until nutritionally rehabilitation was achieved.

Text of Methods has been updated to provide these details as per our response to Comment 1b from Reviewer 1, above.

Reviewer #4, Comment #1: *High-risk deliveries including pre-term or some known maternal characteristics (pre-eclampsia or gestational diabetes) are associated with low birth weight and need to be accounted for in the analyses. If you remove the cases with high-risk deliveries including pre-term birth and low-birth weights whether the differences in intervention groups or primary outcomes are still the same or different.*

Response: With regard to baseline characteristics that might affect outcomes, our expectation would be that the randomisation would distribute these evenly between active vs. control arms, thereby minimising the potential for them to act as confounders. In keeping with this expectation, birthweight was distributed evenly between participants randomised to vitamin D vs. placebo (Table 1). We prefer not to present results of exploratory analyses including adjustment for additional covariates that were not pre-specified in our statistical analysis plan in any published manuscript. However, we are happy to conduct an exploratory analysis to reassure the reviewer that no such confounding operated, by repeating analysis of the primary outcome with additional adjustment for birth weight. When we attempted this exploratory analysis, the random effects model did not converge when additionally correcting for baseline birth weight. However, we were able to obtain an effect estimate using a fixed effects model:

- Without correction for baseline birth weight: adjusted mean difference in WFHZ at 2-month follow-up: 0.02; 95% CI -0.22 to 0.25; P=0.88 (vs. 0.02; 95% CI -0.20 to 0.23; P=0.89 for mixed effects model)
- With correction for baseline birth weight: aMD 0.03; 95% CI -0.21 to 0.26; P=0.83

With regards to the second part of this comment, we conducted a sensitivity analysis excluding participants who had pre-term birth and low birthweight. Effect estimates yielded by this sensitivity analysis are very similar to those yielded by the primary analysis:

Rebuttal Table 2: WFHZ at 2-month follow-up by allocation: overall, and excluding participants who had pre-term birth and low birthweight.

	Timepoint, months from baseline	Placebo: mean value (s.d.), [n]	Vitamin D: mean value (s.d.), [n]	Adjusted mean difference (95% CI) ^[1]
Anthropometric outcomes				
Primary analysis: weight-for-height or -length z-score, all participants	+2	-2.59 (1.09), [128]	-2.58 (1.10), [123]	0.02 (-0.20, 0.23) ^[2]
Exploratory sensitivity analysis: weight-for-height or -length z-score, excluding pre-term and low birth weight	+2	-2.58 (1.09), [76]	-2.54 (1.02), [68]	-0.01 (-0.28, 0.26)

Taken together, results of these exploratory analyses provide reassurance that results of the trial were not confounded by low birthweight.

Reviewer #4, Comment #2. Please provide the references for including the input values in the sample size computation.

Response: reference added to the revised manuscript.

Reviewer #4, Comment #3. In Table 1, there is no reason to report the distribution of baseline characteristics for the overall patients. I would suggest removing this column and reporting the p-value to ensure the randomization is achieved or not.

Response: We prefer the current presentation, but are happy to amend it if the Editor advises that this breaches house style. Formal statistical comparison of baseline characteristics of participants allocated to active vs placebo arms of a trial is not generally encouraged – indeed CONSORT guidelines advise against it (7).

Reviewer #4, Comment #4. In Table 2, please highlight significant p-values.

Response: Our understanding is that this is not ‘house style’ for Nature Communications – but happy to make this change if the Editor advises otherwise.

Reviewer #4, Comment #5. P-values were not reported for faecal inflammatory markers. Also, please do not compute the p-value using a nonparametric test on log-transformed data. Either use a nonparametric test on the original data or use a parametric test on the log-transformed data.

Response: thank you for highlighting this inconsistency. We have revised Table 2 so that it presents results from the analysis of log₁₀-transformed data on faecal inflammatory markers using our generalised mixed model:

Rebuttal Table 3: revised effect estimates now presented Table 2 of revised manuscript

	Timepoint, months from baseline	Placebo: mean value (s.d.), [n]	Vitamin D: mean value (s.d.), [n]	Adjusted mean difference (95% CI) ⁽¹⁾
Faecal inflammatory markers				
Mean log ₁₀ (faecal myeloperoxidase, ng/ml) (s.d.)	+2	7.79 (1.08), [116]	7.65 (1.14), [115]	-0.18 (-0.51, 0.15)
	+6	7.38 (1.01), [113]	7.42 (1.14), [109]	0.03 (-0.31, 0.36)
Mean log ₁₀ (faecal neopterin, nmol/L), (s.d.)	+2	6.45 (1.54), [116]	6.65 (1.50), [115]	0.20 (-0.16, 0.57)
	+6	6.37 (1.63), [114]	6.04 (1.49), [109]	-0.36 (-0.72, 0.00)
Mean log ₁₀ (faecal alpha-1 antitrypsin, mg/dL) (s.d.)	+2	2.60 (1.11), [116]	2.83 (0.78), [115]	0.21 (-0.05, 0.47)
	+6	2.64 (0.87), [114]	2.86 (0.83), [109]	0.21 (-0.04, 0.47)

Reviewer #4, Comment #6. Table 3, extremely unbalanced sample sizes were observed between baseline 25(OH)D ≥ 50 vs. < 50 , and therefore a Welch t-test or nonparametric tests should be preferred without using the random effects models.

Response: We prefer not to abandon our generalised mixed model, which converged despite some imbalance in sub-group sizes. This approach was pre-specified in our statistical analysis plan, so it would not be appropriate to present findings of another approach in the revised manuscript.

However, we are happy to apply Welch t-tests in an exploratory analysis (i.e. not for presentation in the revised manuscript), highlighting that this approach cannot include the adjustments pre-specified in our original model or yield P values for interaction, which are needed to interpret findings of sub-group analyses. Results are presented below.

Rebuttal Table 4: exploratory sub-group analysis showing major efficacy outcomes at 2-month follow-up by allocation and baseline vitamin D status, using Welch t-tests to compare mean values for participants allocated to placebo vs. vitamin D within sub-group strata.

Outcome	Baseline 25(OH)D concentration, nmol/L	Placebo, mean value (s.d.), [n]	Vitamin D, mean value (s.d.), [n]	Adjusted mean difference (95% CI), calculated using Welch t-tests
Anthropometric/Clinical Outcomes				
Weight-for-height/length z-score	<50	-2.66 (1.24), [17]	-1.68 (1.04), [7]	-0.98 (-2.03 to 0.71)
	≥50	-2.74 (1.22), [30]	-2.73 (1.14), [55]	-0.01 (-0.55 to 0.54)
Lean mass index, Ω^{-1}	<50	12.92 (2.26), [17]	13.30 (2.57), [7]	-0.38 (-2.82 to 2.07)
	≥50	12.34 (2.13), [30]	12.60 (2.47), [55]	-0.26 (-1.28 to 0.76)
Mean mid-upper arm circumference, cm	<50	11.24 (1.39), [17]	12.0 (0.50), [7]	-0.76 (-1.56 to 0.03)
	≥50	11.58 (1.07), [30]	11.59 (0.89), [55]	-0.01 (-0.47 to 0.45)
Weight-for-age z-score	<50	-3.56 (1.00), [17]	-2.62 (1.13), [7]	-0.94 (-2.02 to 0.13)
	≥50	-3.71 (1.24), [30]	-3.53 (1.08), [55]	-0.19 (-0.73 to 0.35)
Neurodevelopmental Outcome				
Overall MDAT score	<50	62.53 (26.24), [17]	64.29 (16.83), [7]	-1.76 (-20.52 to 17.00)
	≥50	62.00 (26.41), [30]	59.24 (22.64), [55]	2.76 (-8.68 to 14.21)

Reviewer #4, Comment #7. Mean serum 25(OH)D was only available for 1/2 to 1/3 of the patients in each group. However, safety outcomes of hypervitaminosis D were observed for all participants.

Response: Thank you for drawing our attention to this point. We have revised Table 4 so that denominators are presented for each cell, according to data availability.

Rebuttal Table 3: data presented in Table 4 of revised manuscript, entitled ‘Safety outcomes by allocation’

		Placebo (n=129) ⁽¹⁾		Vitamin D (n=127) ⁽²⁾	
		No. of events	Proportion of participants with one or more events (%)	No. of events	Proportion of participants with one or more events (%)
Fatal serious adverse event ⁽³⁾		2	2/128 (1.6)	0	0/123 (0.0)
Non-fatal serious adverse event ⁽³⁾		9	9/128 (7.0)	3	3/123 (2.4)
Non-serious adverse event leading to discontinuation of study supplement		0	0/128 (0.0)	0	0/123 (0.0)
Relapse of SAM (complicated or uncomplicated)		4	4/128 (3.1)	3	3/123 (2.4)
Other monitored safety conditions	Hypercalcaemia ⁽⁴⁾	6	5/128 (3.9)	2	2/123 (1.6)
	Hypervitaminosis D ⁽⁵⁾	1	1/83 (1.2)	22	22/85 (25.9)
	Hypercalciuria ⁽⁶⁾	56	50/128 (39.1)	51	44/123 (35.8)

1, no. of participants who took at least one dose of placebo and for whom follow-up data were available for at least one timepoint.
 2, no. of participants who took at least one dose of vitamin D and for whom follow-up data were available for at least one timepoint.
 3, causes for each event presented in Table S5, Supplementary Material. 4, defined as serum albumin-adjusted calcium concentration >2.65 mmol/L. 5, defined as 25(OH)D concentration >220 nmol/L. 6, defined as urinary calcium:creatinine molar ratio >1.00.

Reviewer #4, Comment #8. *Due to separate groups in serum 25(OH)D concentrations at 2 months in the active group, it would be worth determining a Spearman rank correlation between serum 25(OH)D concentrations and primary outcomes within each treatment group.*

Response: We pre-specified that we would analyse trial data by intention-to-treat, and not by attained 25(OH)D, since responder analyses of the type proposed are open to bias (participants with larger 25[OH]D responses to supplementation often have different baseline characteristics to those with smaller responses, and these may influence outcomes).

For the purposes of this rebuttal (i.e. not for inclusion in the revised manuscript), we are happy to run an exploratory analysis to determine whether the primary outcome differed between participants in the intervention arm with higher vs. lower 25(OH)D concentrations at 2-month follow-up. Respectfully, we prefer not to use Spearman rank correlations to explore this possibility, as this approach cannot incorporate pre-specified adjustments. A fixed effects model with 2-month WHZ as the dependent variable and 2-month 25(OH)D concentrations (<200nmol/l vs >/=200nmol/L) as the independent variable, correcting for site and limiting to data from participants randomised to the intervention arm yielded the following effect estimate:

Adjusted mean difference 0.26; 95% CI -0.38 to 0.90; P=0.42

In other words, there was no statistically significant difference in 2-month WHZ of participants in the intervention arm of the trial who experienced larger vs. smaller 25(OH)D increments in response to supplementation.

Reviewer #4, Comment #9. *Please follow CONSORT reporting guidelines.*

Response: We have followed CONSORT reporting guidelines throughout, and a CONSORT checklist was included in our original submission. If there are specific aspects where the reviewer or editor feels that guidelines have not been adhered to, we would be happy to look at these.

We hope that the responses above address all concerns raised, and look forward to hearing from you in due course.

Best wishes,

Prof Adrian Martineau

References

1. Sturgeon JP, Tome J, Dumbura C, Majo FD, Ngosa D, Mutasa K, et al. Inflammation and epithelial repair predict mortality, hospital readmission, and growth recovery in complicated severe acute malnutrition. *Sci Transl Med.* 2024;16(736):eadh0673.
2. World Health Organisation. WHO guideline on the prevention and management of wasting and nutritional oedema (acute malnutrition) in infants and children under 5 years. Geneva: World Health Organisation; 2023.

3. Pakistan. Go. Pakistan National Guidelines for Community-based Management of Acute Malnutrition. 2014.
4. Naz S, Hoodbhoy Z, Jaffar A, Kaleem S, Hasan BS, Chowdhury D, et al. Neurodevelopment assessment of small for gestational age children in a community-based cohort from Pakistan. *Arch Dis Child*. 2023;108(4):258-63.
5. Saleem J, Zakar R, Zakar MZ, Belay M, Rowe M, Timms PM, et al. High-dose vitamin D3 in the treatment of severe acute malnutrition: a multicenter double-blind randomized controlled trial. *Am J Clin Nutr*. 2018;107(5):725-33.
6. Institute of Medicine. Dietary Reference Intakes for Calcium and Vitamin D. Washington, DC: National Academy Press; 2011.
7. Schulz KF, Altman DG, Moher D. CONSORT 2010 Statement: updated guidelines for reporting parallel group randomised trials. *BMC medicine*. 2010;8:18.

High-dose vitamin D3 in the treatment of complicated severe acute malnutrition in Pakistan: a double-blind randomised controlled trial (ViDiSAM): NCOMMS-24-44517.

Response to second round of reviewer comments

Reviewer #1, Comment #1: *The point about timing of vitamin D administration was answered in the context of "mirroring" their previous trial. WHO recommendations for the treatment of severe acute malnutrition do NOT preclude administration of needed micronutrients during in-hospital treatment. In fact it is expected standard of care. If the objective here was to improve outcomes in the convalescent phase of treatment of such children, it should be clearly reflected in the title and trial description.*

Response: We have amended the title of the manuscript to highlight our focus on improving outcomes in the convalescent phase of treatment – it now reads ‘High-dose vitamin D3 to improve outcomes in the convalescent phase of complicated severe acute malnutrition in Pakistan: a double-blind randomised controlled trial (ViDiSAM)’.

We have also amended the abstract to specify the timing of administration of trial medication and our intention to improve outcomes in the convalescent phase of treatment, such that the second sentence now reads ‘We conducted a randomised placebo-controlled trial in Lahore, Pakistan, to determine whether two oral doses of 200,000 international units (IU) vitamin D3 (the first administered on or before the day of hospital discharge and the second administered 14 days later) would benefit children aged 6-59 months during the convalescent phase of complicated SAM’. The trial description in Methods already clearly reflects our intention to improve outcomes in the convalescent phase, as evidenced by our choice of primary outcome (‘The primary outcome was the difference between active and placebo participants in mean weight-for-height or -length z-score (WHZ) at 2 months after administration of the first dose of IMP’), and by the clear statement regarding timing of administration of vitamin D (‘the first dose administered on or before the day of the participant’s hospital discharge and the second dose administered 14 days later, with a tolerance of 10-24 days.’)

Reviewer #1, Comment #2: *The issue with the laboratory measurement challenges (mislabeling) now explained, does raise questions on trial quality and should be mentioned clearly in the trial description. I appreciate the information provided on the subset studied and would suggest that this information be included in the paper. Moving the analysis of outcomes by vit D status to supplementary material is prudent.*

Response: To address this comment, we have added the information on the subset of participants for whom baseline serum 25(OH)D measurements were available (new Table S6, Supplementary Material) along with the following wording to the trial description to address this comment: ‘Sub-group analyses by baseline vitamin D status were conducted in a subset of 106 participants for whom both baseline serum 25(OH)D concentrations and outcome measures were available: P values for interaction for all of these sub-group analyses were also >0.05 (Table S5, Supplementary Material). Baseline characteristics of participants included in this sub-group analysis were similar to those of trial participants overall (Table S6, Supplementary Material).’

Reviewer #1, Comment #3: Comment 3 also pertained to vit D status in children available from the National Nutrition Survey 201, and comparison to the values found in the trial among severely

malnourished children. Have the authors considered this point in terms of external validity of their findings and subjects?

Response: Prevalence of vitamin D deficiency defined at the 50 nmol/L (20 ng/mL) threshold was 62.7% among children aged 6-59 months in the National Nutrition survey, as compared with 63.4% among children in the current study. The fact that these figures are similar does not raise concerns re external validity. The following text has been added to the Discussion: 'Prevalence of vitamin D deficiency (defined as serum 25[OH]D concentration <50 nmol/L) among study participants at baseline (63.4%) was similar to that reported among children aged 6-59 months in the 2018 Pakistan National Nutritional Survey (62.7%) (1), supporting the external validity of our findings.'

We hope that the responses above address all concerns raised, and look forward to hearing from you in due course.

Best wishes,

Prof Adrian Martineau

References

1. Pakistan Ministry of National Health Services Regulations and Coordination. National Nutrition Survey 2018: Key Findings Report. Government of Pakistan; 2018.